# Modelling convective cell lifecycles with a copula-based approach

Chien-Yu Tseng[1], Li-Pen Wang[1,2], and Christian Onof[2]

[1]National Taiwan University, Taipei, 106319, Taiwan
[2]Imperial College London, London, SW7 2AZ, United Kingdom

**Correspondence:** Li-Pen Wang (lpwang@ntu.edu.tw)

**Abstract.** This study proposes an algorithm designed to model convective cell lifecycles, for the purpose of improving the representation of convective storms in rainfall modelling and forecasting. We propose to explicitly model cell property inter-dependence and temporal evolution. To develop the algorithm, 165 effective convective storm events occurring between 2005 and 2017 in Birmingham, UK, were selected. A state-of-the-art storm tracking algorithm was employed to reconstruct convec-

tive cell lifecycles within each selected event. The investigation of these cell lifecycles proceeded in three stages. The initial stage involved statistically characterising individual properties of convective cells, including rainfall intensity, spatial extent at peaks, and lifespan. Subsequently, an examination of the inter-correlations amongst these properties was conducted. In the final stage, the focus was on examining the evolution of these cell properties during their lifetimes. We found that the growth and decay rates of cell properties are correlated with the cell properties themselves. Hence the need to incorporate this correlation

structure into the process of sampling convective cells. To resolve the dependence structures within convective cell evolution, a novel algorithm based on vine copulas is proposed. We show the proposed algorithm's ability to sample cell lifecycles preserving both observed individual cell properties and their dependence structures. To enhance the algorithm's applicability, it is linked to an exponential shape model to synthesise spatial fields for each individual convective cell. This defines a model which can readily be incorporated into rainfall generators and forecasting tools.

## 1 Introduction

Climate change has emerged as an urgent environmental concern, driving non-negligible changes in global weather patterns, particularly the frequency and intensity of extreme events (Trenberth et al., 2003; Liu et al., 2009; Guhathakurta et al., 2011). A notable trend linked to this phenomenon is the intensification of localised, short-duration rainfall extremes, often attributed to severe convective systems (Guerreiro et al., 2018; Fowler et al., 2021b, a; Lenderink et al., 2021). This trend highlights

the need to enhance the modeling of convective storms within climate models and to better account for their impacts on the subsequent hydrological applications. Accordingly, there has been a growing incorporation of convection-permitting (CP) models in climate research (Trapp et al., 2019; Coppola et al., 2020; Prein et al., 2020; Halladay et al., 2023; Archer et al., 2024). Despite the demonstrated capability to enhance the simulation of extreme precipitation events, there are still deficiencies in CP models. For example, these models often tend to produce overly intense heavy rainfall, and their reliability is heavily

influenced by the quality of lateral boundary forcing (Prein et al., 2017; Kendon et al., 2021). Moreover, their extensive

computational requirements, particularly when a large number of ensemble members are needed, may have hindered their widespread adoption (Kendon et al., 2021; Lucas-Picher et al., 2021a).

A suitable alternative to the generation of high-resolution rainfall simulations is stochastic rainfall/weather generators. These generators are mostly data-driven, constructed with historical data over the area of the interest. Since they can well reflect the regional statistical features, they have been increasingly utilised in many hydrological applications, such as urban drainage design (Willems, 2001; Thorndahl and Andersen, 2021) and flood risk assessment (Simões et al., 2015; Diederen and Liu, 2020; Wright et al., 2020).

Based on the data used for model construction, these generators can be roughly categorised into two types. The first type relies solely on rainfall data, including measurements from rain gauges and weather radars over the study area. Amongst these, point process-based models are widely used. These models simulate the rainfall process using two Poisson-cluster processes: the first models storm arrivals, and within each storm, the second process models the arrivals of rain cells (Rodriguez-Iturbe et al., 1987, 1988). Initially, point process-based models were employed mainly for generating rainfall time series (Cowpertwait, 1994; Onof and Wheater, 1994, 1993). They were subsequently extended to spatial-temporal rainfall modeling, utilising either rain gauge data from dense networks (Wheater et al., 2000; Willems, 2001; Koutsoyiannis et al., 2003; Burton et al.; Segond and Onof, 2009) or rainfall properties derived from radar data (Féral et al., 2003; Luini and Capsoni, 2011; McRobie et al., 2013; Muñoz Lopez et al., 2023). Correlated random field generators are another increasingly popular category within the first type. These models typically conduct continuous simulations of space-time correlated rainfall fields, preserving key features of the rainfall process observed in radar data across various spatial and temporal scales. This category includes models that generate rainfall as a nonlinear transformation of Gaussian random fields with parametric covariance forms (Ferraris et al., 2003; Paschalis et al.; Benoit et al., 2018; Wilcox et al., 2021; Papalexiou et al., 2021; Green et al., 2024), as well as models based on multifractal processes (Schertzer and Lovejoy, 1987; Gires et al., 2020).

The second type of generators can be technically similar to or extended from the first type. However, instead of using solely rainfall data, they further incorporate additional weather variables, such as temperature, humidity and wind, to generate more comprehensive simulations of weather conditions (Wilks and Wilby, 1999; Ivanov et al., 2007; Jones et al., 2010; Fatichi et al., 2011; Peleg and Morin, 2014; Peleg et al., 2017; Papalexiou, 2018). These models aim to capture the dependencies between rainfall processes and other weather variables, enabling the generation of internally consistent weather scenarios for impact studies and climate-related assessments (Sparks et al.; Ahn, 2020; Van De Velde et al., 2023).

In spite of significant advancements in the development of storm and weather generators, with the increasing demand for high-resolution rainfall modelling (Ochoa-Rodriguez et al., 2015; Eggimann et al., 2017) –whether the purpose is for subsequent hydrological applications or for accurately representing intense, short-duration convective storms–, it is still an open challenge to integrate the modelling of convective storms into these generators. Various efforts and strategies have been proposed in the literature to address this issue. Some approaches adopt the concept of typical Point process-based models but calibrate the model parameters exclusively with rain gauge or radar data only from pre-identified convective storm events (Willems, 2001; McRobie et al., 2013; Wilcox et al., 2021).

Alternatively, instead of pre-filtering input data, some approaches modify the model structure to explicitly account for the correlation between storm properties or to differentiate between convective and stratiform rainfall types within stochastic rainfall generators (Jo Kaczmarska and Onof, 2014; Peleg and Morin, 2014; Zhao et al., 2019; Onof and Wang, 2020; Muñoz Lopez et al., 2023). For example, Jo Kaczmarska and Onof (2014) and Onof and Wang (2020) introduced an additional parameter to a randomized Bartlett-Lewis rectangular pulse model to associate rain cell intensity with storm duration. This

adjustment has significantly improved the model's ability to reproduce extreme rainfall properties at sub-hourly and hourly timescales. Similarly, Peleg and Morin (2014) incorporated a convective rain cell generator module into the process of rainfall generation. This module, based on HYCELL –a Gaussian-Exponential hybrid rain cell shape model (Féral et al., 2003)–, is specifically responsible for sampling convective rain cells. This incorporation enables the proposed weather generator to effectively reproduce the observed spatial structure of rainfall over the study area.

Despite the progress made in modeling convective storms, most convective generators do not explicitly account for cell evolution. For example, the generators proposed by McRobie et al. (2013) and Peleg and Morin (2014) model only the occurrence frequency and advection of convective cells, neglecting the evolution—an important characteristic of convective cells highlighted by Rigo and Carmen Llasat (2016). Ignoring this evolution may lead to misrepresentations of rainfall extremes and subsequent hydrological responses (Muñoz Lopez et al., 2023). Although it is critical to model cell evolution, to the authors'

knowledge, only a limited number of generators in the literature explicitly model this process (Paschalis et al.; Ghirardin et al., 2016; Wilcox et al., 2021). For example, Wilcox et al. (2021) proposed temporally disaggregating convective rainfall using a predefined hyetograph pattern, consisting of a symmetrical triangular peak representing the convective front of the storm, followed by a stratiform tail of lower intensity. Likewise, Ghirardin et al. (2016) represented convective cell evolution with a predefined exponential function, associating the size of a rain cell with its peak intensity.

As can be seen, in most existing evolution methods, the modelling of a cell's temporal profile (or lifecycle) is largely simplified, and inter-correlation amongst cell properties throughout the lifecycle is not satisfactorily accounted for. To properly address these deficiencies, in this work, we propose to develop an algorithm that enables the stochastic generation of not only convective cells but also their lifecycles, allowing for explicit modeling of cell evolution.

    This paper is organised as follows. In Sect. 2, we provide descriptions of the pilot domain and rainfall datasets utilised in this

study. Section 3 explains the proposed methodology for developing a stochastic convective cell lifecycle algorithm, including techniques for extracting convective cell lifecycles from high-resolution radar data, statistical characterisation of the properties of cell lifecycles and their inter-dependence, and the copula models employed to model these properties and inter-dependence. Section 4 evaluates the capacity of the proposed algorithm to preserve the statistical properties of observed convective cells. Finally, in Sect. 5, we summarise the key findings from this work and discuss potential further developments and applications

of the proposed algorithm.

## 2 Dataset

### 2.1 Pilot domain

The pilot site under study in this paper is located near Minworth (see the area with dark purple shading in Fig. 1 (right), approximately $30 \times 34$ km$^2$), a highly urbanised catchment located in the West Midlands region in England. This catchment covers the entire Birmingham city (the area with light pink shading in Figure 1 (right)) and a significant portion of the industrial Black Country area. With a population of around 1.5 million people, it spans an area of 431 km$^2$. Minworth was selected as the pilot site due to its extensive history of surface water flooding, primarily attributed to localised, high-intensity convective storms during the summer months. A recent assessment conducted by the Birmingham City Council estimated that approximately 22,900 homes in the area are susceptible to this type of flooding (Birmingham City Council, 2015).

Centred at the Minworth catchment, two domains with different sizes were used in this study to perform storm cell tracking and the storm cell properties analysis, respectively. These domains are:

- **Tracking domain** is a $500 \times 500$ km$^2$ region centred at Minworth (the squared area with a black dashed border line in Fig. 1 (left)). The storm tracking was conducted using radar images over this domain. The size of this domain was chosen to ensure a reliable estimate of storm motion (Wang et al., 2015).

- **Analysing domain** is a $250 \times 250$ km$^2$ region centred at Minworth. After the storm tracking process, only rain cells with centroids located within this domain were used for further analyses. There are two main reasons for choosing this domain. Firstly, as shown in Fig. 1 (left), this domain has a higher density of radar data sites, resulting in better data quality. Secondly, selecting this domain helps reduce the impact of boundary effects on the extracted rain cell data. In some cases, storm cells may only partially enter the tracking domain during a given event period. Including the properties of these partially tracked cells in the analysis may lead to faulty estimate of their statistical behaviors.

### 2.2 Radar rainfall data

The pilot domain falls within the coverage area of C-band radars operated by the Met Office (UKMO) (see Fig. 1 (left)). These radars operate at a frequency of 5.6 – 5.65 GHz, with an operational range of approximately 200 km, a range resolution of 600 m, and a beam width of 1°. Considering the distances of the pilot catchments from the radar (approximately 50 km), the 1°beam width results in an angular resolution of approximately 870 m. The UKMO radars conduct scans at 5 different elevations (0.5°, 1.0°, 2.0°, 3.0°, and 4.0°) within a scan repeat cycle of 5 minutes. During the period of 2005 – 2017, the radar covering the pilot catchment primarily had single-polarisation capabilities. Although the UKMO radars are being upgraded to include dual-polarisation and Doppler capabilities (Darlington et al., 2016), radar Quantitative Precipitation Estimates (QPEs) at the UKMO are still derived based on single polarisation data.

Radar QPEs for the pilot catchment were obtained from the British Atmospheric Data Centre (BADC). These estimates were provided at spatial and temporal resolutions of 1 km and 5 minutes, respectively. They correspond to a quality-controlled multi-radar composite product generated using the UK Met Office Nimrod system (Golding, 1998). This Nimrod product incor-

porates various corrections to address inherent errors in radar rainfall measurements. These corrections include identification and removal of anomalous propagation (e.g., beam blockage and clutter interference), attenuation correction, vertical profile correction, and an hourly-based nationwide mean field bias correction (Harrison et al., 2000, 2009; Sandford, 2015).

## 2.3   Event selection

The event selection in this study was first conducted using ground rain gauge records over the Minworth catchment to determine durations for storm events. Based on these durations, the Met Office C-band Nimrod radar rainfall data was then used for convective cell extraction. The event selection criteria were designed based on the WaPUG (Wastewater Planninig Users Group) standard. This has provided guidance for best practice in urban drainage management in the UK since 1984. Though WaPUG has now been replaced by the CIWEM (Chartered Institution of Water and Environmental Management) Urban Drainage Group (UDG), its guidelines for selecting storm events for the calibration and verification of urban drainage models remain relevant. Specifically, we referred to User Note 06 (Use of Rainfall Data from Flow Surveys) produced by WaPUG in 2009 (Gooch, 2009). While a new rainfall modelling guide was published in 2016 (CIWEN, 2016), the principles for event selection are similar. These principles include criteria for event durations, cumulative rainfall, instantaneous rainfall rates, and the quality of rainfall data.

For our study area, the general criteria are instantaneous rainfall rates greater than 5 mm/h and cumulative rainfall greater than 5 mm, ensuring effective rainfall and subsequent runoff. Since our focus is on convective cell lifecycle modelling, we specifically chose events between May and July, filtering out those without any 5-min rainfall intensity greater than 5.6 mm/h (equivalent to a reflectivity of 35 dBZ –a threshold commonly used to identify convective regions according to the Marshall-Palmer relationship (Marshall and Palmer, 1948)), as well as those with durations shorter than 15 minutes. In addition, we excluded events with consecutive periods of missing radar data, resulting in a total of 165 events between 2005 and 2017 summer times (on average 12.7 events per year).

A summary of these events is given in Table 1. This includes the number of selected events, as well as statistics for storm durations (in hours), areal and maximum rainfall event totals (in mm), and 5-min areal and maximum peak rainfall intensities (in mm/h) for each year of the study period. The areal and maximum totals represent, respectively, the spatial average and the highest individual pixel value of cumulative rainfall over the catchment area for each storm event. Similarly, the areal and maximum peaks denote the spatial average and the highest individual pixel value of instantaneous rainfall intensity. As seen, the average durations of the selected events are mostly shorter than 20 hours, with the longest ones exceeding 70 hours, likely comprising several convective systems separated by durations that are however shorter than 6 hours.

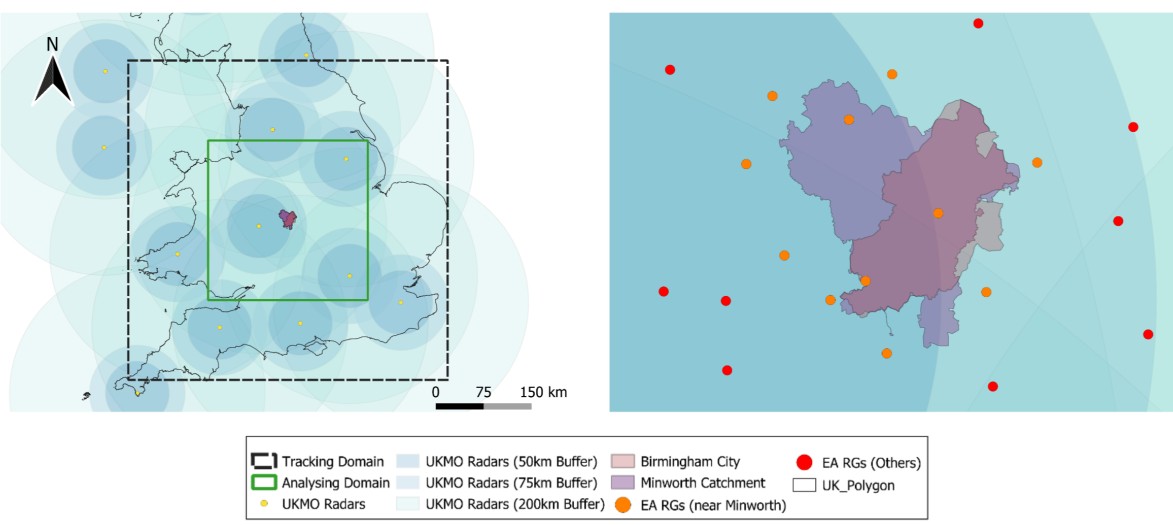

**Figure 1.** Pilot catchment, study domains and rainfall monitoring networks. Left: the Minworth catchment (in the very middle of the map), the neighbouring UKMO radar sites and their effective coverages, and the tracking and analysing domains are illustrated. Right panel: a close view of the Minworth catchment (and its relative location to the city of Birmingham), and the neighbouring Environment Agency (EA) rain gauges.

**Table 1.** Statistical summary of the selected convective storm events between 2005 and 2017 summer months (May to July).

| Year | No. of events | Ave. Duration (h) | Max. Duration (h) | Min. Duration (h) | Ave. Areal Total (mm) | Ave. Max. Total (mm) | 5-min Ave. Areal Peak (mm/h) | 5-min Ave. Max. Peak (mm/h) |
|------|------|-------|-------|------|-------|-------|------|-------|
| 2005 | 10 | 17.85 | 29.33 | 3.58 | 13.42 | 21.92 | 8.49 | 36.44 |
| 2006 | 8  | 21.84 | 54.75 | 2.58 | 7.69  | 12.90 | 5.87 | 27.12 |
| 2007 | 18 | 20.62 | 52.08 | 1.50 | 16.23 | 24.53 | 7.02 | 29.43 |
| 2008 | 9  | 23.23 | 70.58 | 0.33 | 10.78 | 16.89 | 10.10 | 56.00 |
| 2009 | 23 | 16.77 | 50.50 | 3.08 | 9.98  | 16.07 | 5.97 | 29.16 |
| 2010 | 11 | 12.77 | 38.17 | 1.92 | 7.72  | 14.05 | 5.42 | 25.66 |
| 2011 | 12 | 14.60 | 37.92 | 3.58 | 5.88  | 10.00 | 5.47 | 22.38 |
| 2012 | 14 | 19.23 | 61.92 | 3.25 | 14.04 | 22.10 | 8.35 | 27.47 |
| 2013 | 9  | 12.48 | 30.83 | 3.17 | 12.13 | 17.13 | 9.91 | 32.15 |
| 2014 | 12 | 13.47 | 31.75 | 1.42 | 12.13 | 18.03 | 9.99 | 31.30 |
| 2015 | 15 | 17.03 | 84.83 | 0.58 | 8.15  | 15.07 | 8.39 | 40.36 |
| 2016 | 12 | 14.71 | 75.83 | 2.67 | 9.75  | 20.88 | 6.63 | 36.19 |
| 2017 | 12 | 22.26 | 36.08 | 4.42 | 8.57  | 12.53 | 4.53 | 17.94 |

## 3 Methodology

### 3.1 Overview

The lifecycle of a convective cell typically consists of three stages. These are developing, mature and dissipating (Kim et al., 2012). In between these stages, cell merging and/or splitting may occur, further complicating the cell evolution process (Rigo and Carmen Llasat, 2016). In this study, we focus upon developing an algorithm that samples single-core convective cell lifecycles, without considering cell merging and splitting events. The proposed methodology for developing such an algorithm is illustrated in Figure 2.

The process starts with the application of an object-based storm tracking algorithm to high-resolution radar images, which allows us to extract convective cells and their temporal associations (tracks). Next, individual cell lifecycles are reconstructed from the extracted cells and tracks. A conceptual model representing the convective cell lifecycle is then proposed, and relevant properties are computed and statistically characterised from the extracted lifecycle data. It is important to note that, to better characterise the cell evolution process, we not only characterise the distributions of individual properties (e.g. major and minor axes and peak intensity of cells) but also analyse their inter-dependence.

Finally, based on the statistical features of the lifecycles, a copula-based algorithm is developed to stochastically generate convective cell lifecycles. This algorithm enables the generation of synthetic lifecycles that preserve the observed statistical properties and inter-dependence of convective cell properties.

The proposed methodology comprises three main steps. These are:

- **Cell lifecycle extraction:** Initially, high-resolution radar images are processed using Enhanced TITAN, a state-of-the-art convective cell tracking algorithm proposed by Muñoz et al. (2018). This algorithm identifies convective cells and establishes temporal associations between cells across successive time steps. Then, the algorithm proposed by Cheng et al. (2024) is applied to the output of the Enhanced TITAN cell tracking to extract individual cell lifecycles.

- **Statistical characterisation of cell lifecycles:** This step involves a comprehensive examination of selected lifecycle properties. Not only individual cell characteristics such as intensities, lifespans, and spatial extents, but also the temporal evolution of these properties, in terms of growth and decay rates over their lifespans, are investigated. Moreover, the dependence structures amongst these properties are assessed, and the possibility of modelling these structures using copulas is explored.

- **Copula-based stochastic convective cell generation:** Finally, a copula-based algorithm is proposed for generating stochastic convective cell lifecycles. This algorithm reproduces cell lifecycles while preserving their observed characteristics and interdependencies. Moreover, it can be enhanced by incorporating an exponential rain cell shape model known as MultiEXCELL (Luini and Capsoni, 2011), enabling the generation of convective cell lifecycles with spatially-distributed convective cells.

A detailed explanation of each of these steps is given in the following sections.

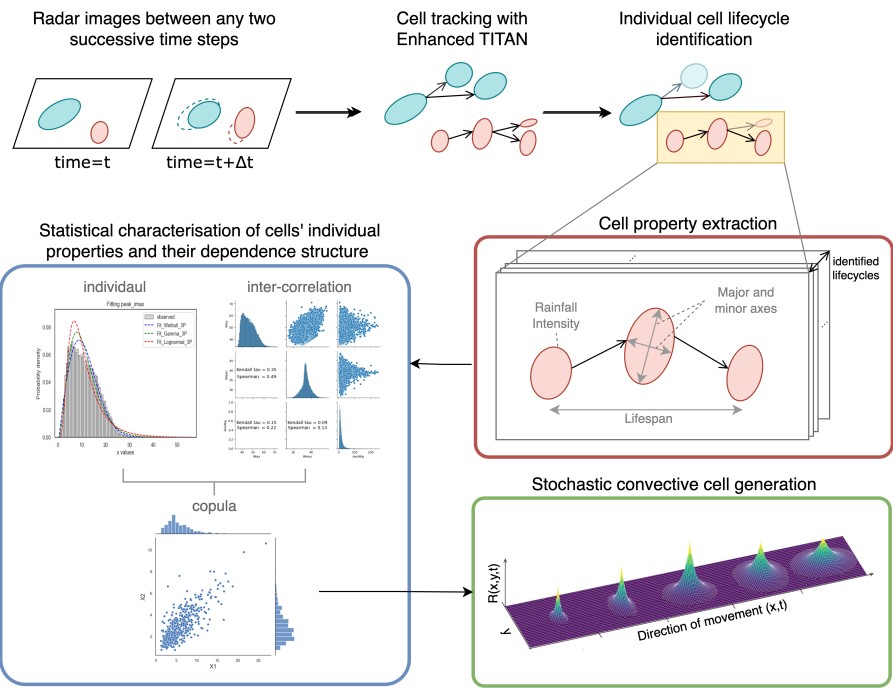

**Figure 2.** Overview of the proposed methodology. It comprises three main steps: 1) cell property extraction from high-resolution radar images, 2) Statistical characterisation of cells' individual properties and their dependence structure, and 3) an algorithm enables stochastic generation of convective cell lifecycles.

## 3.2  Cell lifecycle extraction

A two-stage approach, adopted from the method proposed by Cheng et al. (2024), was utilised here to extract and to construct convective cell lifecycles from high-resolution radar images. The first stage of this approach is to identify convective cells and to establish their temporal associations (hereafter, "tracks") between any two successive time steps. Here, a state-of-the-art storm cell tracking algorithm, named enhanced TITAN, was employed (Muñoz et al., 2018). The second stage is to post process the extracted cells and tracks, such that individual cell lifecycles can be further isolated from the outcome of the first stage. Here, an algorithm, based upon the graph theory, was utilised. The details of the key techniques of these two stages are explained as follows.

### 3.2.1  Stage 1: Convective cell tracking over high-resolution radar images

The enhanced TITAN storm tracking algorithm, proposed by Muñoz et al. (2018), was employed here to facilitate the identification and tracking of convective cells from high-resolution radar images. This algorithm was developed based upon the

widely-used TITAN (Thunderstorm Identification, Tracking, Analysis and Nowcasting; see Dixon and Wiener (1993)) algorithm and was specifically tailored to work with high-resolution radar images. Similarly to TITAN, the enhanced TITAN comprises two main steps: 1) cell identification and 2) temporal association. The former identifies individual convective cells from each radar image, and the latter establishes temporal relationships between cells from successive time steps. There are however some noticeable deficiencies in the original TITAN, which hinder its applicability to tracking, for example, in the case of small-sized but high-intensity or fast-moving cells. Specific treatments are thus introduced in the enhanced TITAN to improve these deficiencies. These include:

- **A multi-threshold segmentation (MTS) method**: this method is inspired by the hierarchical threshold segmentation (HTS) method proposed by Peak and Tag (1994). Whereas the original TITAN used a single threshold for cell identification, enhanced TITAN applied a range of threshold values (e.g. ranging from 35 to 40 dBZ) to the separation of storm cells from each radar image. This results in a number of cell clusters with different sizes from each radar image. Each of these clusters is then modelled as a tree-like data structure with a given number of hierarchies, of which each node at a given tree level represents an isolated rainfall region filtered by a given threshold. A heuristic pruning process, which considers cell sizes and shapes, is then performed to 'trim' cells from the tree, such that individual cells from each radar image can be identified. For the details of the pruning algorithm, readers are referred to Appendix 1 of Muñoz et al. (2018).

- **Incorporation of field motion tracker**: One of the main challenges in object-based storm tracking is to (temporally) associate those cells identified between successive time steps. The original TITAN, as well as many of its variants, rely on a 'matching' process that minimises an objective function quantifying the level of similarity between any two cells. This optimisation process however does not account for the spatial consistency of the movements between neighbouring cells. In contrast, field-based storm tracking works with entire rainfall fields and tends to derive spatially consistent (or smoother) motion fields. The resulting motion fields however could be overly smooth, failing to capture the deviation of movements of individual (convective) cells from those of neighbouring cells. To overcome the inherent limitations of each type of tracking method, Muñoz et al. (2018) employed a 'hybrid' approach. It first conducts a field-based tracking to obtain a spatially consistent motion field, which provides an initial guess of the movements of each convective cell. Then, the typical 'matching' process in object-based tracking is conducted, With help from field-based tracking: the solution space for the optimisation problem can be effectively reduced, and consequently, the accuracy and efficiency of convective cell tracking can be improved.

This tracking stage yields sets of convective cells and tracks that link cells between successive time steps. These entities however do not conform to a simple list-like structure; instead, the links between cells form a graph-like structure due to the dynamic nature of convective systems, involving frequent merging and splitting of cells. While this graph-like structure is not suitable for extracting statistical properties of individual lifecycles, Cheng et al. (2024) addressed this issue by adopting the method proposed by Liu et al. (2016) to decompose each graph of convective cells and tracks into several individual lifecycles. Details of this method are provided in the following section.

### 3.2.2 Stage 2: Extraction of individual cell lifecycles

In graph theory, a graph data structure is made of sets of vertices (or nodes) connected by edges (or links). By specifying the starting and termination vertices, one can identify a path meeting specific criteria (e.g., shortest distance). Building upon the approach introduced by Liu et al. (2016), Cheng et al. (2024) applied this graph theory concept to extract the 'most representative' lifecycle from a cluster of interconnected rain cells. Initially, they modelled each cell cluster as a directed spatial-temporal graph, where vertices represented cells and directed edges depicted tracks linking successive cells. Then, Cheng et al. (2024) specified those cells lacking previous cells as starting vertices and those lacking subsequent cells as termination vertices. Paths with the minimal variation in mean intensities between each pair of starting and termination vertices can be identified.

To achieve this, a weight $W_{ij}$ was assigned to a track associating cells $i$ and $j$, formulated based on an estimate derived from mean reflectivity values of cells $i$ and $j$. Specifically,

$$W_{ij} = |R_i - R_j| \tag{1}$$

where $R_i$ is the normalised mean reflectivity of a given cell $i$, computed as

$$R_i = \frac{I_i - I_{min}}{I_{max} - I_{min}} \tag{2}$$

Here, $I_{\min}$ and $I_{\max}$ denote the minimum and maximum mean reflectivity of the cell cluster to which cells $i$ and $j$ belong.

The weight $W_{ij}$ thus indicates the level of difference in mean reflectivity estimates between two successive cells. Utilising these weights, the Dijkstra's shortest path algorithm was applied to identify all possible shortest paths between any two starting and termination vertices (Dijkstra, 1959). Finally, the path with the longest length among these identified paths was extracted as the most representative lifecycle from the cell cluster under analysis. For a comprehensive understanding of the above lifecycle extraction algorithm, readers are referred to Appendix A in Cheng et al. (2024).

### 3.3 Conceptualising convective cell lifecycles

A conceptual model, based on the patterns observed in our extracted cell lifecycles and those in the literature (Kim et al., 2012; Shusse et al., 2015; Rigo and Carmen Llasat, 2016), is proposed to model essential cell properties (e.g., intensity and size) as they evolve over time. As illustrated in Fig. 3, each cycle undergoes growth, peak and decay stages (also known as development, maturity and dissipation stages, respectively), characterised by three key attributes. These are :

- **Lifespan** is computed as the duration ($D_{\mathrm{L}}$) from the initial to the last available cell.

- **Peak** consists of maximum reflectivity ($I_{\mathrm{max,\ peak}}$, in dBZ) and lengths of major and minor axes ($S_{\mathrm{maj,\ peak}}$ and $S_{\mathrm{min,\ peak}}$, in km) at the peak time step.

- **Temporal variation** includes the growth and decay rates ($R_{\cdot,\mathrm{growth}}$ and $R_{\cdot,\mathrm{decay}}$) of the selected properties (e.g., maximum reflectivity and major and minor axes). These rates are calculated as the ratios of peak values to initial (or last) values.

This estimation indicates a linear variation in property values during growth and decay periods. This estimation assumes a linear progression of property values during growth and decay periods, a trend supported by previous research (Capsoni and Luini, 2009; Rigo and Carmen Llasat, 2016).

Subsequently, ten cell properties are required to model the geometric and intensity variations throughout lifecycles (see Table 2 for a summary). To comply with the requirement of the proposed conceptual model, we further filtered out those lifecycles shorter than three time steps and those with multiple-peak patterns from the lifecycles obtained from the extraction process (detailed in Sect. 3.2) since no clear growth, peak and decay stages could be identified. This enables focusing analysis on distinct evolutionary trends. Subsequently, a total of 27,731 cell lifecycles will be used for further statistical analysis of their

characteristics.

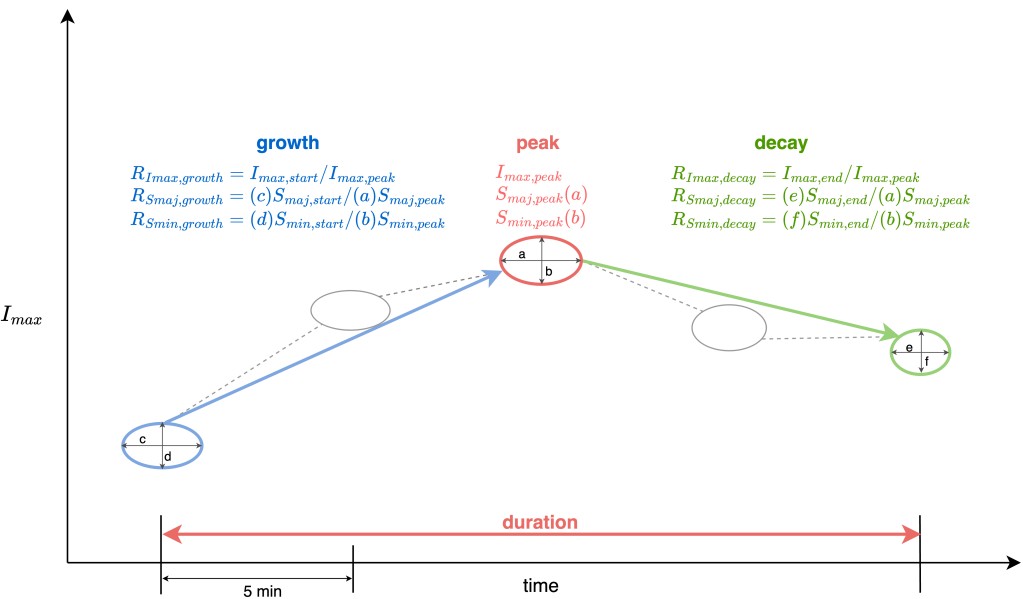

**Figure 3.** Schematic for conceptualising convective cell lifecycle.

## 3.4 Statistical characterisation of cell lifecycles

To analyse the statistical features of lifecycles, here we first focus on characterising cell properties individually, then on the inter-dependence between them. The details of the characterisation are explained as follows.

### 3.4.1 Individual property characterisation

Our first step involves identifying the optimal distribution for each property individually. This process is relatively straightforward. Firstly, we pre-selected a set of candidate distributions based on visual inspection and existing literature. Subsequently,

we employed the maximum likelihood estimation (MLE) method to fit the distribution parameters. The AIC (Akaike information criterion) that accounts for both fitting likelihood estimates and model complexity was then used to determine the most appropriate distribution for each property. The fitting results of the peak properties using pre-selected distributions are shown

in Fig. 4, where the corresponding AICs obtained from each candidate distribution are given. In addition, a summary in Table 2 highlights the most suitable distribution for each cell property along with the associated parameters. Remarkably, the best-fit distribution for each peak property generally aligns with those identified in prior research, such as McRobie et al. (2013).

**Table 2.** Summary of key properties to conceptualise the lifecycle model and the corresponding optimal probability distribution and parameters.

| Property | Description | Fitted distribution | Distribution parameters | AIC |
|---|---|---|---|---|
| Duration | | | | |
| $D_{\mathrm{L}}$ | Total time duration of the cycle (5-min intervals) | Exponential | $\lambda=0.239, \gamma=3.000$ | 1.33E+05 |
| Peak | | | | |
| $I_{\mathrm{max,peak}}$ | Maximum intensity at peak (km) | Weibull | $\alpha=12.029, \beta=1.972, \gamma=35.528$ | 1.71E+05 |
| $S_{\mathrm{maj,peak}}$ | Major axis length at peak (km) | Loglogistic | $\alpha=9.807, \beta=2.261, \gamma=3.216$ | 1.93E+05 |
| $S_{\mathrm{min,peak}}$ | Minor axis length at peak (km) | Loglogistic | $\alpha=4.524, \beta=2.599, \gamma=1.915$ | 1.43E+05 |
| Temporal variation | | | | |
| $R_{I\mathrm{max,growth}}$ | The ratio of the initial to the peak maximum intensity (dbz/dbz) | Beta | $\alpha=13.373, \beta=1.329$ | -7.93E+04 |
| $R_{I\mathrm{max,decay}}$ | The ratio of the last to the peak maximum intensity (dbz/dbz) | Beta | $\alpha=10.636, \beta=1.44$ | -6.53E+04 |
| $R_{S\mathrm{maj,growth}}$ | The ratio of the initial to peak major axis length (km/km) | Gamma | $\alpha=0.255, \beta=3.074, \gamma=0$ | 2.75E+04 |
| $R_{S\mathrm{maj,decay}}$ | Ratio of the last to peak major axis length (km/km) | Weibull | $\alpha=0.830, \beta=1.654, \gamma=0.016$ | 2.95E+04 |
| $R_{S\mathrm{min,growth}}$ | Ratio of the initial to peak minor axis length (km/km) | Weibull | $\alpha=0.901, \beta=2.374, \gamma=0$ | 1.94E+04 |
| $R_{S\mathrm{min,decay}}$ | Ratio of the last to peak minor axis length (km/km) | Weibull | $\alpha=0.857, \beta=2.132, \gamma=0$ | 2.19E+04 |

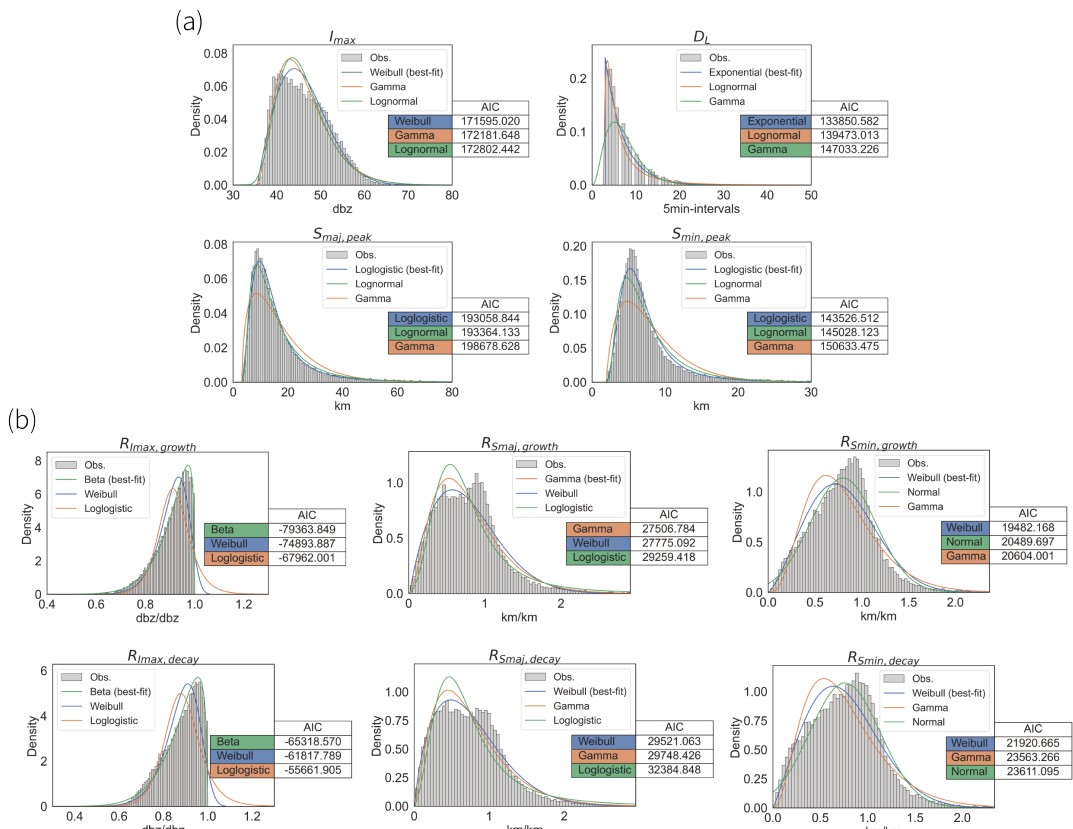

**Figure 4.** Histogram of selected cell properties at peaks (a) and the corresponding growth/decay rates (b) fitted probability distributions. $I_{\text{max, peak}}$: Maximum intensity (in dBZ), $D_L$: cell lifespans (in time step), $S_{\text{maj, peak}}$, $S_{\text{min, peak}}$: lengths of major and minor cell extent (in km), $R_{.,\text{growth}}$: growth rates of selected cell properties, and $R_{.,\text{decay}}$: decay rates of selected cell properties.

### 3.4.2 Dependence structure characterisation

We then turn our attention to exploring the statistical dependence amongst these cell properties. The importance of considering
dependence between different storm properties has been highlighted in numerous studies (e.g. Gyasi-Agyei and Melching, Salvadori and De Michele (2004) and Salvadori and De Michele (2006)). Particularly, it plays a key role of preserving fine-scale statistical features in the rainfall process (Jo Kaczmarska and Onof, 2014; Onof and Wang, 2020). Modelling this dependence however poses a challenge especially as it requires multivariate analysis, which becomes even more complex when the data does not follow a Gaussian distribution.
To address this challenge, we propose to use copula theory for the numerical modelling of the dependence between storm properties. The key idea behind copula theory is to transform the marginal distributions of individual variables into a standard uniform distribution over the interval $[0, 1]$ (Genest and Favre, 2007; Jaworski et al., 2013; Czado, 2019; Tootoonchi et al., 2022). This transformation maintains the dependence structure between the variables while decoupling it from the individual

characteristics of the marginal distributions. We then utilise Spearman and Kendall's $\tau$ correlation coefficients, which are better suited for capturing nonlinear relationships as compared to Pearson's correlation coefficient, to quantify the dependencies between the transformed variables. As depicted in Fig. 5 (a), all selected properties appear to be correlated with each other, with varying levels of dependence. Notably, the correlation coefficients between major and minor axis lengths and those between peak intensity and lifecycle duration are higher than those between other properties. This is consistent with findings in the literature (Willems, 2001; Luini and Capsoni, 2011; McRobie et al., 2013; Muñoz Lopez et al., 2023). In addition, we investigate the dependence between each peak property and the associated growth and decay rates (see Fig. 5 (b)-(d)). To the authors' knowledge, this dependence is often overlooked in the literature. Nevertheless, our analyses suggest a significant dependence between them, with the estimated correlation coefficients are statistically significant at a 0.05 level (95% confidence interval), with p-values less than 0.05, which should not be ignored.

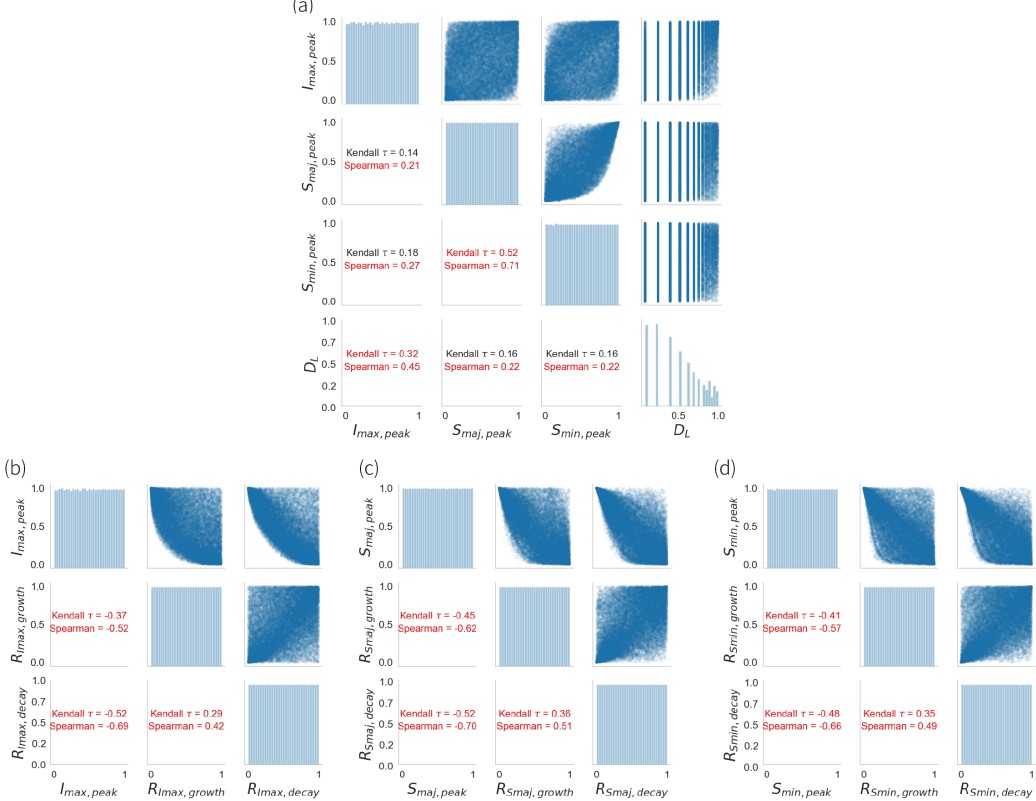

**Figure 5.** Correlation analyses amongst selected cell properties: (a) between peak properties; (b)-(d) between each peak property and the associated growth and decay rates. Kendall's $\tau$ values are displayed in red when the correlation is greater than 0.2.

To model these dependencies, we employ vine copulas, known for their flexibility in capturing various dependence structures. Originating from Joe (1994) and formally defined by Cooke (1997), vine copulas decompose joint distributions into a number of bivariate copulas arranged in a tree-like structure. Each node represents a variable modelled as a univariate random variable, while each edge captures the conditional dependence between variables. However, applying vine copulas to multivariate analysis becomes increasingly complex as the number of variables grows, requiring careful determination of the copula structure and dependence order (Czado and Nagler, 2022). Here, we utilised the Python package `pyvinecopulib` to facilitate the determination of the optimal vine-copula model (Nagler and Vatter, 2023). This process involves two key steps. First, it requires determining the aforementioned tree-like structure for the vine copula. Second, it has to identify the most suitable bivariate copula for each edge of the tree and estimate the model parameters. In this study, following the general practice in the literature, similarly to the identification of optimal distributions for individual properties, we employed AIC to determine the optimal vine copula for the characterisation of the cell property dependence (Tosunoglu et al.; Czado and Nagler, 2022; Latif and Simonovic). A detailed explanation of the bivariate copula family selection, parameter estimation, and the three fitting strategies employed is provided in Supplement S1.

Here, we summarise the selection result. The final selection of the copula structure, dependence order and the bivariate copula for each edge, and the associated AIC values are summarised in Table 3. A 4-dimensional (4D) 2-3-1-4 D-vine copula is used to model the dependence amongst cell duration and peak properties (i.e. $C_{\text{peak}}$), while 3D C-vine copulas are used to model that between each given peak property and its corresponding growth and decay rates (i.e. $C_{I\text{max}}$, $C_{S\text{maj}}$ and $C_{S\text{min}}$). The use of D-Vine for peak properties is due to the lack of a clear dominating variable in the dependence structure amongst the four selected cell properties; thus, the dependence between most pairs of variables has to be modelled. Our D-Vine model comprises three levels of connected trees, where the edges from a lower level constitute the nodes at a higher level.

As illustrated in Fig. 6 (left), at the first level, cell properties are modelled with three bivariate copulas (i.e., 2,3: between $S_{\text{maj,peak}}$ and $S_{\text{min,peak}}$; 3,1: between $S_{\text{min,peak}}$ and $I_{\text{max,peak}}$; and 1,4: between $I_{\text{max,peak}}$ and $D_{\text{L}}$). Based on our analysis, the nonparametric transformation local likelihood kernel estimator (denoted TLL) appears to be the most suitable model. At the second level, the dependence is further modelled on the constraint sets formed by the union of a given pair of variables at the first level (i.e., between 2,1 and 3,1 and between 3,1 and 1,4). Likewise, our analysis suggests using the (non-parametric) TLL copula to characterise these dependence structures. Finally, at the third level, the dependence is modelled between two conditional constraint sets (i.e. 2,1|3 and 3,1|4); here, a parametric Clayton-Gumbel (denoted BB1) copula is found to be the most suitable model.

Unlike the dependence amongst the four selected cell properties, the relationship between each peak property and its corresponding growth and decay rates is primarily influenced by the peak properties themselves. Thus, a 3D C-vine, which typically features a dominant variable conditioning the other variables, is employed to model their dependence structures.

As illustrated in Fig. 6 (right), the peak property (indexed as 1) serves as the dominant variable, and the bivariate dependence structures between this variable and the associated growth (indexed as 2) and decay rates (indexed as 3) are initially modelled individually at the first level. Subsequently, the dependence between two constraint sets (i.e., 2,1 and 3,1) is modelled. In all three dependence structures, the (non-parametric) TLL copula is identified as the most suitable model.

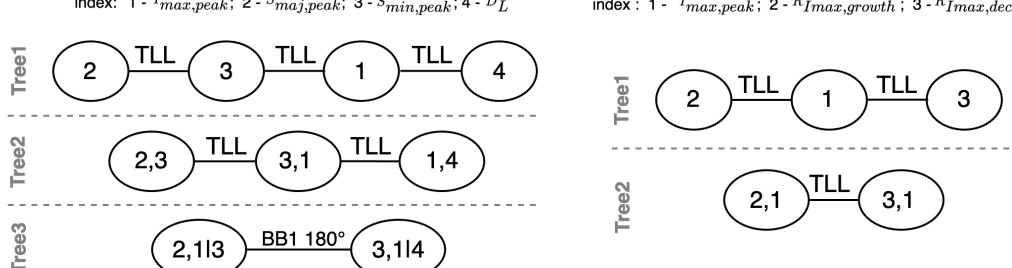

**Figure 6.** Illustration of $C_{peak}$ as D-Vine structure (left), and $C_{I\max}$ as C-Vine structure (right).

**Table 3.** The vine structure of the fitted models with the related paired variables and the value of criteria.

| | Vine Structure | Paired Variable | | Copula Family | Parameter | AIC | Log Likelihood |
|---|---|---|---|---|---|---|---|
| | | | 2,3 | | | | |
| | | Tree1 | 3,1 | | | | |
| $C_{peak}$ | 2-3-1-4 | | 1,4 | TLL | - | -37362.08 | 18893.49 |
| | (D-Vine) | | 2,1\|3 | | | | |
| | | Tree2 | 3,4\|1 | | | | |
| | | Tree3 | 2,4\|1,3 | BB1 180° | 0.037, 1.048 | | |
| $C_{I\max}$ | | | 2,1 | | | -45177.39 | 23116.05 |
| ———— | | Tree1 | | | | ———— | |
| $C_{S\mathrm{maj}}$ | 2-3-1 | | 3,1 | TLL | - | -40262.75 | 20269.11 |
| | (C-Vine) | | | | | | |
| ———— | | | ———— | | | ———— | |
| $C_{S\min}$ | | Tree2 | 2,3 \| 1 | | | -42427.90 | 21352.89 |

### 3.5 Generating convective cell lifecycle with copula

Based on the findings from the statistical characterisation, we propose a copula-based algorithm for the stochastic generation of convective cell lifecycles while preserving observed statistical characteristics and interdependence. As illustrated in Fig.7, the proposed algorithm comprises three steps. First, the four selected cell properties (i.e., three peak properties and cell lifespan) are sampled using a 4D vine copula model. Next, each sampled peak property is used to conditionally sample the correspond-

ing growth and decay rates using a 3D copula. The cell properties and their variations along the lifespan can be therefore

reconstructed. Finally, the EXCELL model is employed to further generate spatially-distributed rainfall intensities for cells with the sampled properties. It is important to note that this final step of linking to the EXCELL model is optional and serves to demonstrate the flexibility of the proposed algorithm to integrate with existing models.

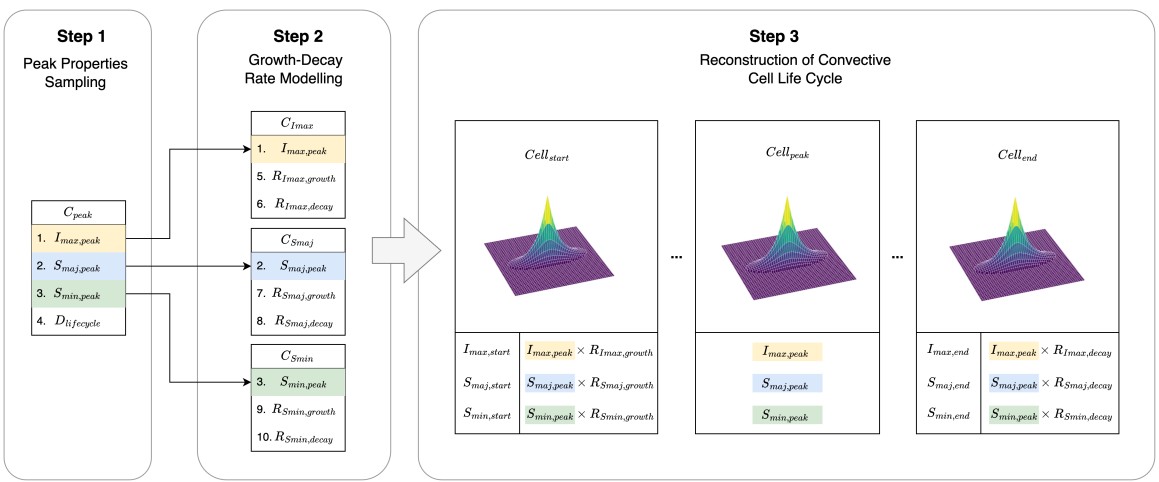

**Figure 7.** Overview of the proposed copula-based algorithm for generating convective cell lifecycles.

There are two main technical challenges of implementing this algorithm. Firstly, conditional sampling from a multi-dimensional copula. Particularly, at step 2, we have to conduct conditional sampling from a 3D copula. This process requires efficient algo-

345 rithms due to the complexity of handling dependencies. Secondly, constructing spatially-distributed rain cells at Step 3 involves synthesising spatially-distributed patterns while ensuring consistency with the sampled properties.

For the former challenge, part of the conditional simulation method proposed by Aas et al. (2021) is utilised. This method was originally developed to estimate individual contributions, in terms of Shapley values (Štrumbelj and Kononenko, 2010), from a number of mutually correlated *features* to the performance of a given machine learning model, where the dependence

between features was modelled with vine copulas. Here, instead of features, we adjust the algorithm to work with a given cell peak property and the corresponding growth and decay rates.

Consider the conditional sampling of growth and decay rates given a peak property $q_1$ as an example. Here, $q_1$ represents the quantile of a randomly sampled value $u_1$, ranging between 0 and 1, obtained from the 4D D-vine copula in Step 1. Firstly, we construct a vector $\boldsymbol{u}$ containing $u_1$ and two additional values randomly sampled from an independent uniform

distribution $U[0,1]$. Thus, $\boldsymbol{u} = (u_1, u_2, u_3)$. Next, we perform the inverse Rosenblatt transformation to convert $\boldsymbol{u}$ into another vector $\boldsymbol{v} = (v_1, v_2, v_3)$, where $v_j$ corresponds to $u_j$ $(j = 1, 2, 3)$. In the case of the 3D C-vine copulas utilised in this study, this inverse operation is termed:

$$v_1 = F^{-1}(u_1), v_2 = F^{-1}(u_2|u_1), v_3 = F^{-1}(u_3|u_1, u_2) \tag{3}$$

Here, $F$ represents the 3D C-vine copula modeling the dependence structure between a given peak property (indexed as 1) and the corresponding growth (indexed as 2) and decay (indexed as 3) rates (see Fig. 6 (right)). According to Rosenblatt (1952), for any copula $F$, if $\boldsymbol{u}$ comprises independent random variables, $\boldsymbol{v}$ will follow the distribution defined by copula $F$. Finally, the growth and decay rates conditioned on a given peak property value $q_1$ can be derived via the corresponding (empirical or theoretical) quantile functions. That is, $q_2 = F_2^{-1}(v_2)$ and $q_3 = F_3^{-1}(v_3)$.

For the latter challenge of synthesising spatially-distributed rainfall intensities (in $\mathrm{mm/h}$) within a cell, we utilise a straightforward model called EXCELL (i.e. exponential cell) to generate the spatial pattern with known cell properties (Capsoni et al., 1987). This model employs an exponential decay function to simulate an elliptical rain cell, with rainfall intensity decreasing from a maximum value at the center of the cell. The rain intensity field of a cell centered at the origin without any orientation is represented by the equation:

$$I(i,j) = I_{\mathrm{max},p}\exp\left[-\left(\frac{i^2}{a^2} + \frac{j^2}{b^2}\right)^{0.5}\right] \tag{4}$$

where $I(i,j)$ denotes the rain intensity at each point location along the x-y plane over the cell's domain, $I_{\mathrm{max},p}$ represents the 'point' maximum rain intensity, and $a$ and $b$ are decay rates along the cell's major and minor axes, respectively. In our setting, $a$ and $b$ are set to $S_{\mathrm{maj}}$ and $S_{\mathrm{min}}$, respectively, and the rainfall intensity decreases to $I_{\mathrm{max},p}/e$ at the edge of the cell area.

However, the $I_{\mathrm{max}}$ values used in this study are derived from 1-km radar imagery, representing areal-averaged rainfall intensities over a $1 \times 1 \ \mathrm{km}^2$ pixel, rather than point rainfall intensities. Therefore, a relationship must be established to convert $I_{\mathrm{max}}$ (areal-average) to $I_{\mathrm{max},p}$ (point rainfall intensity). To achieve this, we integrate Eq. 4 over a circular domain with a 1-km diameter centered on the cell. This yields the following approximate relationship:

$$I_{\mathrm{max},p} = \frac{2I_{\mathrm{max}}}{\pi S_{\mathrm{maj}}S_{\mathrm{min}}\left[1 - (r_1 - 1)\exp(-r_1)\right]} \tag{5}$$

where $r_1 = 1/2S_{\mathrm{maj}}$. This formula enables the conversion of the areal-averaged radar rainfall intensity, $I_{\mathrm{max}}$ into the point rainfall intensity, $I_{\mathrm{max},p}$.

Here, we present an example of using the EXCELL model to generate spatially-distributed convective cells for a lifecycle randomly sampled from the proposed algorithm. Figure 8 (a) shows the cell structures of this 6-time-step lifecycle in a 3D view, with corresponding cell properties at each time step. For this specific sample, the intensity peak occurs at the third time step, and the growth and decay of maximum rainfall intensities throughout the lifecycle are clearly visible. In addition, we provide a plan view of the sampled lifecycle in Fig. 8(b), highlighting the temporal variation in the major and minor axes.

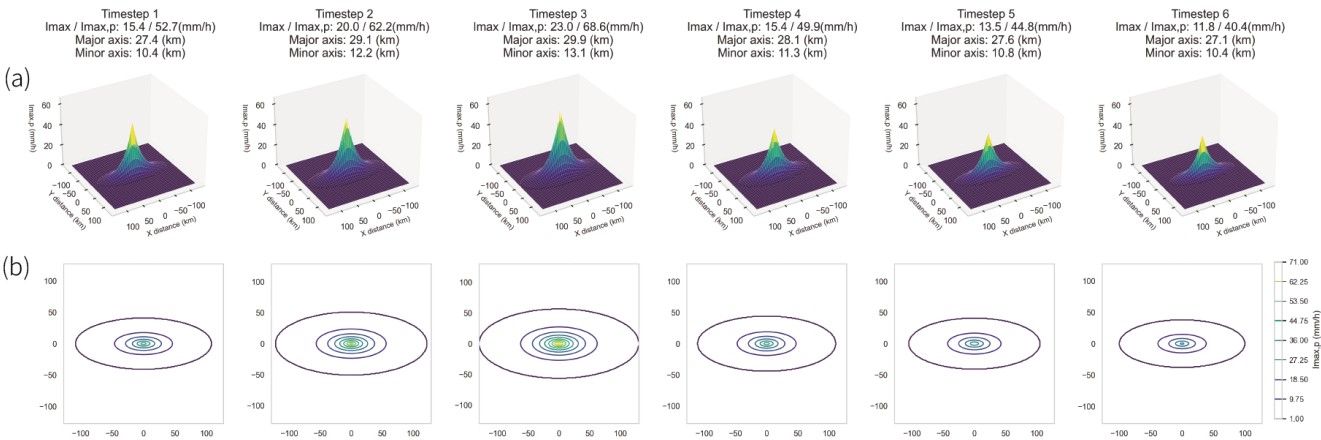

**Figure 8.** Illustration of a convective cell lifecycle generated using the EXCELL model, sampled from the proposed algorithm.

It is worth noting that the EXCELL model generates rather idealised and simplistic spatial patterns for rain cells. Several variants or similar (yet more complex) models have been proposed in the literature to generate statistically more intricate or visually realistic patterns (Féral et al., 2003; von Hardenberg et al., 2003; Rebora and Ferraris, 2006; Luini and Capsoni, 2011; Peleg and Morin, 2014). However, common findings from these works suggest that characterising rain cells with an exponential model is adequate for many applications.

In addition to the technical challenges, there is a critical consideration when conducting sampling from copulas. This involves the method used for sampling, which can be empirical or theoretical. Empirical sampling derives quantiles from the empirical distribution formed by the original data, while theoretical sampling derives quantiles from the fitted probability distributions. The former can better preserve the observed frequency, whilst the latter can infer unseen estimates. This difference in sampling methods may impact the sampled properties, an aspect we will discuss in Sect. 4.

## 4    Results and discussion

### 4.1    Evaluation method and metrics

In this section, we will explore the results from evaluating the proposed algorithm for generating convective cell lifecycle samples. Our primary focus is to assess whether the observed dependence structure amongst peak properties and cell duration, as well as their individual statistical properties, can be well preserved. In addition, we will infer the cell properties for the first and the last time steps of each lifecycle from the sampled peak properties and their corresponding growth and decay rates, and then determine if the observed properties for the first and the last time steps are maintained in the inferred ones.

To evaluate the ability to replicate the observed dependence, we employ Kendall's $\tau$ to compute correlation coefficients amongst sampled (or simulated) properties and compare them with the observed correlations. Note that Kendall's $\tau$ is a non-

parametric measure of the strength and direction of the correlation between two variables. It ranges from -1 to 1, with values closer to -1 (or 1) indicating a stronger negative (or positive) correlation.

Furthermore, to evaluate the similarity between observed and sampled (or simulated) marginal distributions, we utilise Perkins' skill score. This score quantifies the common area between any two probability density functions (PDFs), defined as:

$$S_{\text{score}} = \sum_{i=1}^{n} \min(Z_{s_i}, Z_{o_i}) \tag{6}$$

where $S_{\text{score}}$ represents the skill score, $Z_s$ and $Z_o$ denote the frequencies of the simulated and observed values in a given bin, respectively, and $n$ is the number of bins used to calculate the corresponding PDF (Perkins et al.). The Perkins' skill score, ranging from 0 to 1, is a measure of agreement between simulated and observed distributions, with 1 indicating a perfect match and 0 suggesting no overlap. A higher skill score implies a better agreement between the simulated and observed marginal distributions of cell properties. In practice, a skill score greater than 0.8 is considered excellent, suggesting a strong match between the simulated and observed distributions.

In the following sections, we will discuss the findings derived from 100 ensembles of 27,000 lifecycle samples, where the number of samples at each ensemble member is approximately equal to the number of extracted cell lifecycles.

## 4.2 Preservation of statistical properties at peaks

Here, we begin by examining the statistical characteristics of individual cell properties at peaks. We present results from an arbitrary ensemble member of 27,000 samples in Fig. 9 (a), demonstrating distributions derived from both empirical (blue line, denoted Sim $(C^e)$) and theoretical (green line, denoted Sim $(C^t)$) sampling processes. In the empirical approach, copula-space samples [0,1] are converted into the property's value space via empirical quantile functions. In contrast, the theoretical method uses quantile functions based on fitted probability distributions. As can be seen, both processes exhibit $S_{\text{score}}$ values exceeding 0.9, indicating strong agreement between observed and simulated probability density functions (PDFs). While empirical sampling more closely replicates the observed PDFs, theoretical sampling covers a broader range of values, suggesting it can sample unobserved property values.

Similar trends are observed in the distributions of $S_{\text{score}}$ values across 100 ensemble members (see Fig. 9 (b)). Empirical sampling generally yields slightly higher $S_{\text{score}}$ values with more concentrated distributions, while theoretical sampling produces slightly lower $S_{\text{score}}$ values but with distributions spread over a larger range. Nonetheless, both sampling methods demonstrate satisfactory preservation of individual statistical properties.

In terms of modelling the dependence structure at peaks, we begin by presenting the results from an arbitrary ensemble member of peak properties sampling (see Fig. 10). To demonstrate the impact of utilising the proposed copula model, we highlight two well-known dependencies: between peak rainfall intensity and lifecycle duration (i.e. $I_{\text{max,peak}}$ vs. $D_{\text{L}}$, Fig. 10 (a)) and between major and minor axis lengths of cells (i.e. $S_{\text{maj,peak}}$ vs. $S_{\text{min, peak}}$, Fig. 10 (b)). The left columns in both Figures 10 (a) and (b) show results with copula incorporated, while the right columns display results without it. To better visualise the

quality of dependence modelling, we provide the dependence structures in two spaces: the original variable space (in the upper row) and the $[0,1] \times [0,1]$ copula space (in the lower row).

It is evident that while the marginal distributions can be accurately reproduced in both cases, only those sampled with the proposed 4D copula model can effectively preserve the dependence structures amongst peak properties. This is further confirmed by examining the corresponding Kendall's $\tau$ correlations. As given in Fig. 10, samples obtained with the copula exhibit Kendall's $\tau$ correlations comparable to those observed, indicating close alignment with the original data. In contrast, samples generated without the copula show nearly negligible correlations, suggesting a lack of preservation of the underlying dependence structure.

Furthermore, Figure 11 (a) provides a detailed visualisation of the pairwise dependence structures within the 4D copula for an arbitrary ensemble member in the $[0,1] \times [0,1]$ copula space. The accuracy of the reproduced dependencies is further supported by the corresponding Kendall's $\tau$ values, which show deviations smaller than $10^{-2}$ when compared to the observed data. This close agreement confirms that the 4D copula captures the complex interdependencies among peak properties and lifecycle duration.

Moreover, the selection of empirical versus theoretical quantile functions in the copula process also affects how well the dependence structure at peaks is preserved. Figures 12 (a) and (b) demonstrate results of the dependence structures amongst peak properties from an arbitrary ensemble member of lifecycle samples, obtained from empirical (blue markers) and theoretical (green markers) quantile functions, respectively. Overall, the proposed 4D vine copula model can effectively reproduce the observed dependence structure, whether obtained from the empirical (Fig. 12 (a)) or theoretical (Fig. 12 (b)) quantile functions. This is validated by the corresponding Kendall's $\tau$ values, with differences resulting from empirical and theoretical quantile functions all in the order of $10^{-2}$ or smaller. However, upon visual inspection, it is evident that the dependence structure resulting from theoretical quantile conversion (Sim ($C^t$)) displays a more divergent pattern compared to that from empirical quantile conversion (Sim ($C^e$)), as indicated by the presence of more outliers in the plots. Consistent with the findings in sampling individual properties, this suggests that theoretical quantile function has the potential to sample unobserved data points, while the empirical one tends to better preserve the observed structure.

Extending our analysis to the results from 100 ensemble members, we computed Kendall's $\tau$ correlations amongst peak properties for each ensemble member. As illustrated in Fig. 13 (a), the distributions of Kendall's $\tau$ correlations amongst peak properties are well centred around the observed Kendall's $\tau$ correlations (red dashed lines) with nearly negligible biases. Those obtained from the empirical quantile functions may slightly outperform those from the theoretical ones, in terms of the level of central tendency, but difference is really insignificant. This reaffirms the proposed algorithm's ability to consistently reproduce the observed dependence structures amongst peak properties.

## 4.3 Capturing lifecycle evolution

Building upon the results from peak property sampling, we can further explore the capability of the proposed algorithm to sample the temporal variations, conditioned on the sampled peak properties.

Similarly to the analysis conducted at peaks, we initially assess the performance in reproducing the observed (individual) marginal distributions, as well as the dependence structures between a given peak property and the corresponding growth

and decay rates. As illustrated in Fig. 14 (a), a strong agreement between the observed and the sampled PDFs is evident. Furthermore, the distributions of Perkins skill scores computed from all 100 ensemble members, resulting from both empirical and theoretical quantile functions, are all highly concentrated and well above 0.9 skill score (see Fig. 14 (b)). This indicates that the proposed algorithm consistently reproduces observed distributions of growth and decay rates for selected properties.

We then shift our attention to the dependence structure between specific peak properties and the corresponding growth and

475 decay rates. As illustrated in Fig. 11 (b)-(d), the proposed 3D copula models effectively preserve the observed correlations between the sampled peak properties and the associated growth and decay rates. This trend holds true for both theoretical and empirical quantile conversions, as displayed in Fig. 12 (c)-(h). In addition, distributions of the Kendall's $\tau$ correlations computed from all 100 ensemble members are well centered around the observed Kendall's $\tau$ values, whether resulting from empirical or theoretical quantile functions (see Fig. 13 (b)-(d)).

Building upon the sampled peak properties and the corresponding growth and decay rates, we can further infer the properties at any stage within a given cell lifecycle. To evaluate the result of the inference, here we first derive the properties for the first and the last time steps of the lifecycles from the sampled peak properties and the corresponding growth and decay rates. We then compare them with the observed properties. As shown in Fig. 15, the distributions of the inferred properties are highly consistent with the observed ones, whether those obtained from the empirical or theoretical quantile functions. This suggests

that the observed properties for the first and the last time steps of the lifecycles are well preserved. This result further reassures that the proposed algorithm can effectively replicate the variations in the temporal profiles of cell lifecycles.

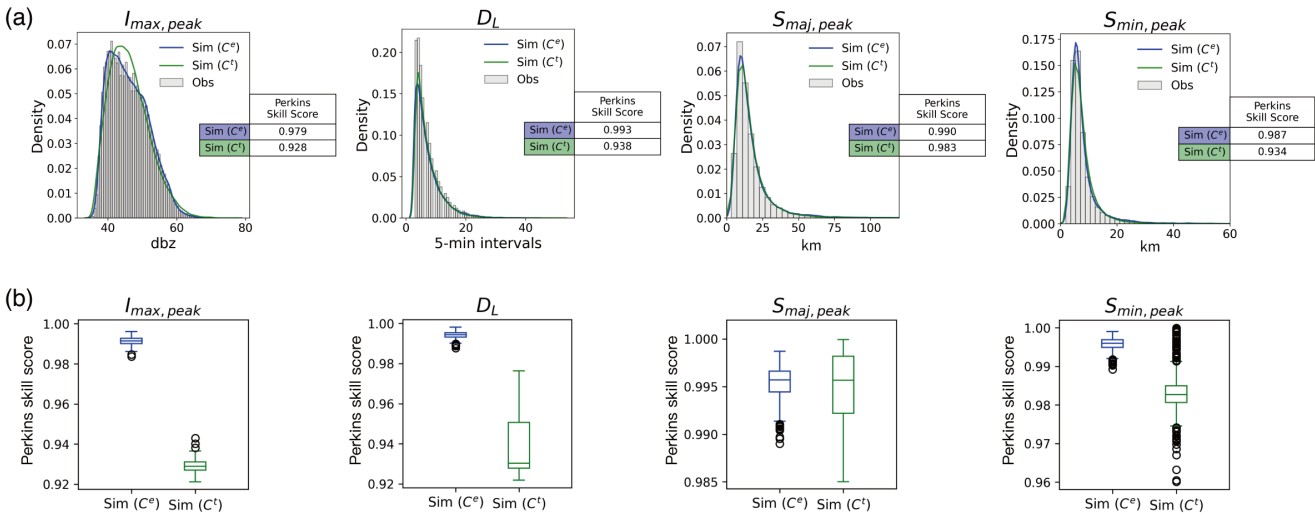

**Figure 9.** Comparisons of marginal distributions derived from empirical quantile conversion (($C^e$))) and theoretical quantile conversion (($C^t$))) sampling process for observed and simulated peak properties: (a) marginal distributions of an arbitrary ensemble member, and (b) distributions of Perkins skill scores derived from 100 ensemble members.

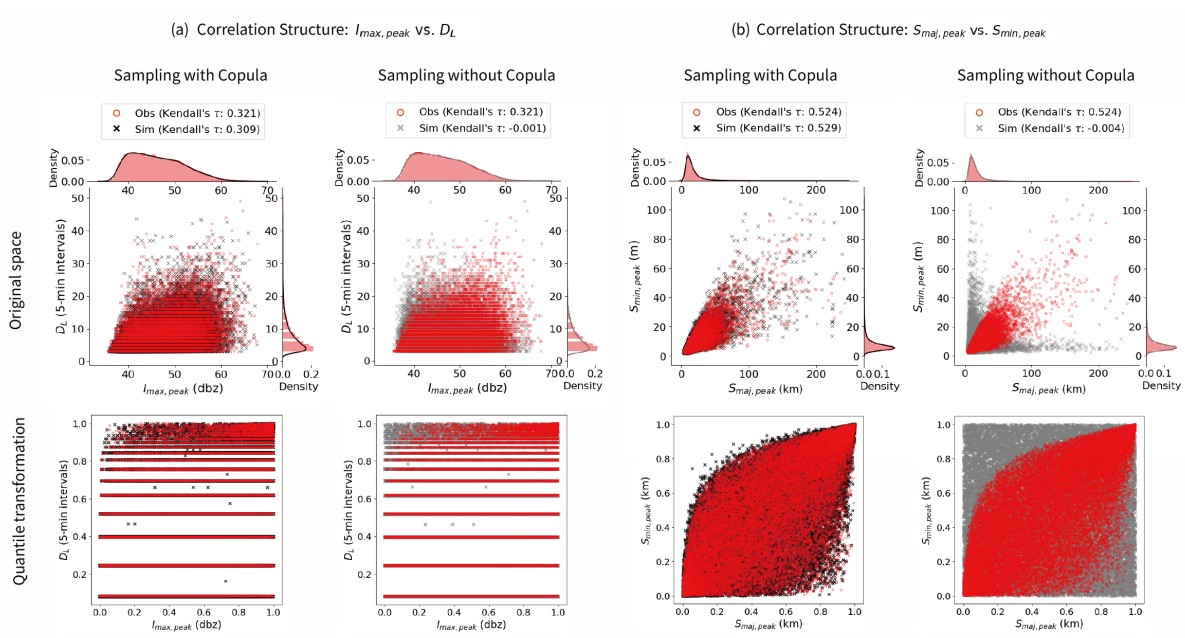

**Figure 10.** A comparison of dependence structures obtained from the observed (red circles) and simulated cell lifecycle samples: (a) $I_{max,peak}$ vs. $D_L$ and (b) $S_{maj,peak}$ vs. $S_{min,peak}$. The left column in (a) and (b) presents results incorporating copula modelling (black crosses: simulated), and the right column shows results without copula modelling (grey crosses: simulated). The upper row displays dependence structure in the original variable space, while the lower row shows after applying the quantile transformation.

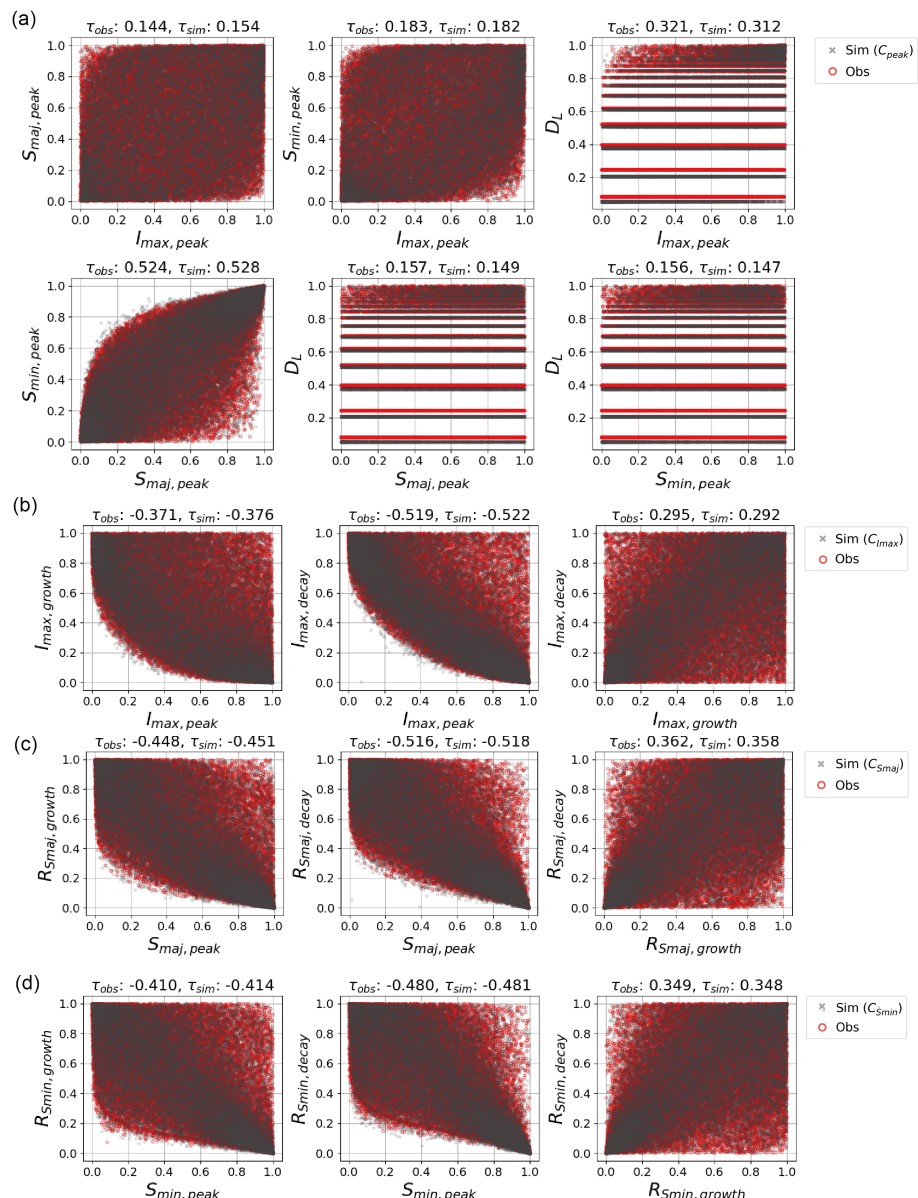

**Figure 11.** Comparisons of the dependence structure between observed (red round markers) and simulated (black crosses) properties obtained from an arbitrary ensemble member. From top to bottom, each row represents results derived from a specific copula model ($C_{peak}$, $C_{Imax}$, $C_{Smaj}$, and $C_{Smin}$).

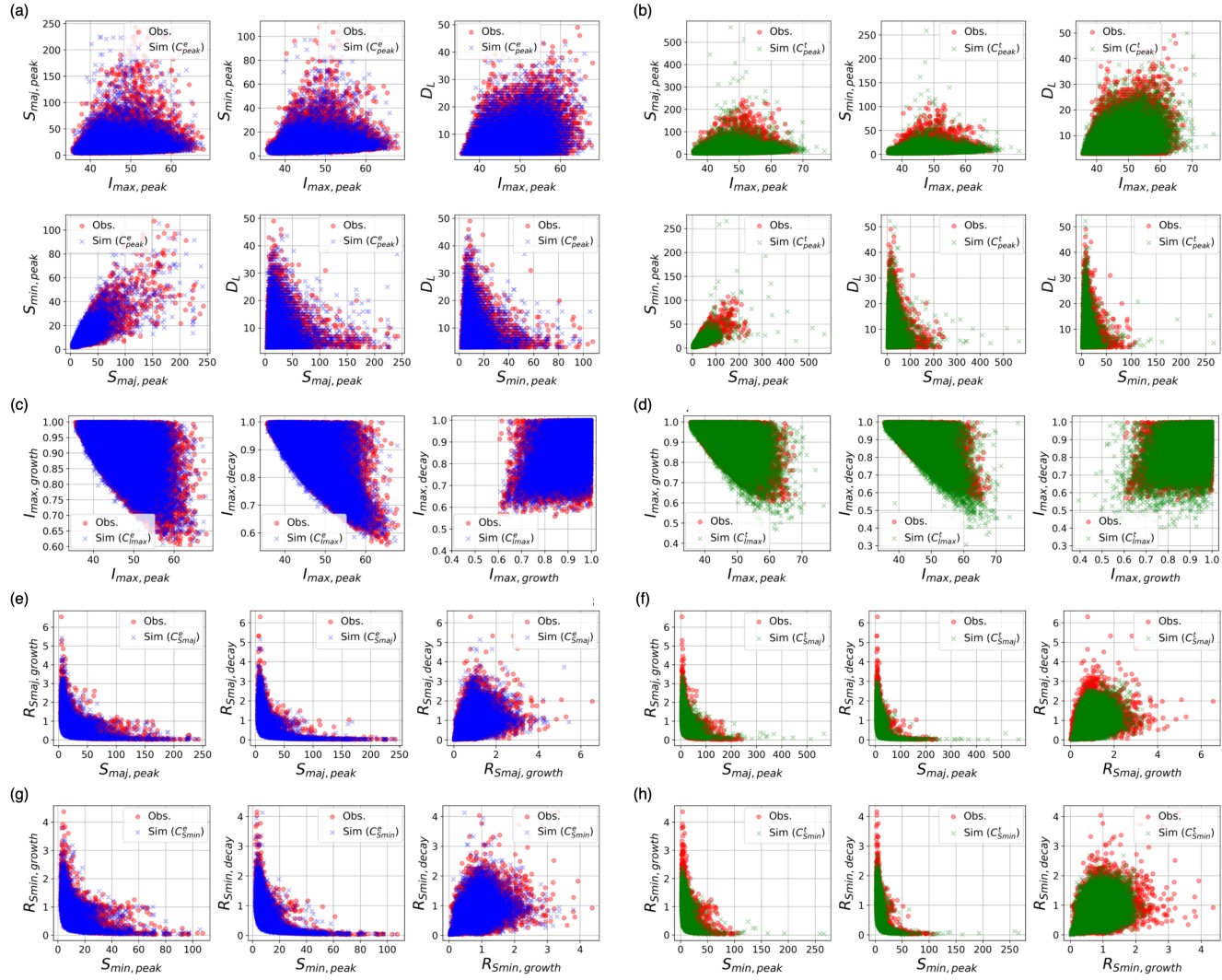

**Figure 12.** Comparisons of the dependence structure between observed (red round markers) and simulated properties in the original value space, using the same ensemble member as in Figure 11. Left three columns of plots (blue markers, denoted $C^e_\cdot$) show simulations derived from empirical quantile functions, whilst right three columns of plots (green markers, denoted $C^t$) show simulations from theoretical quantile functions simulations.

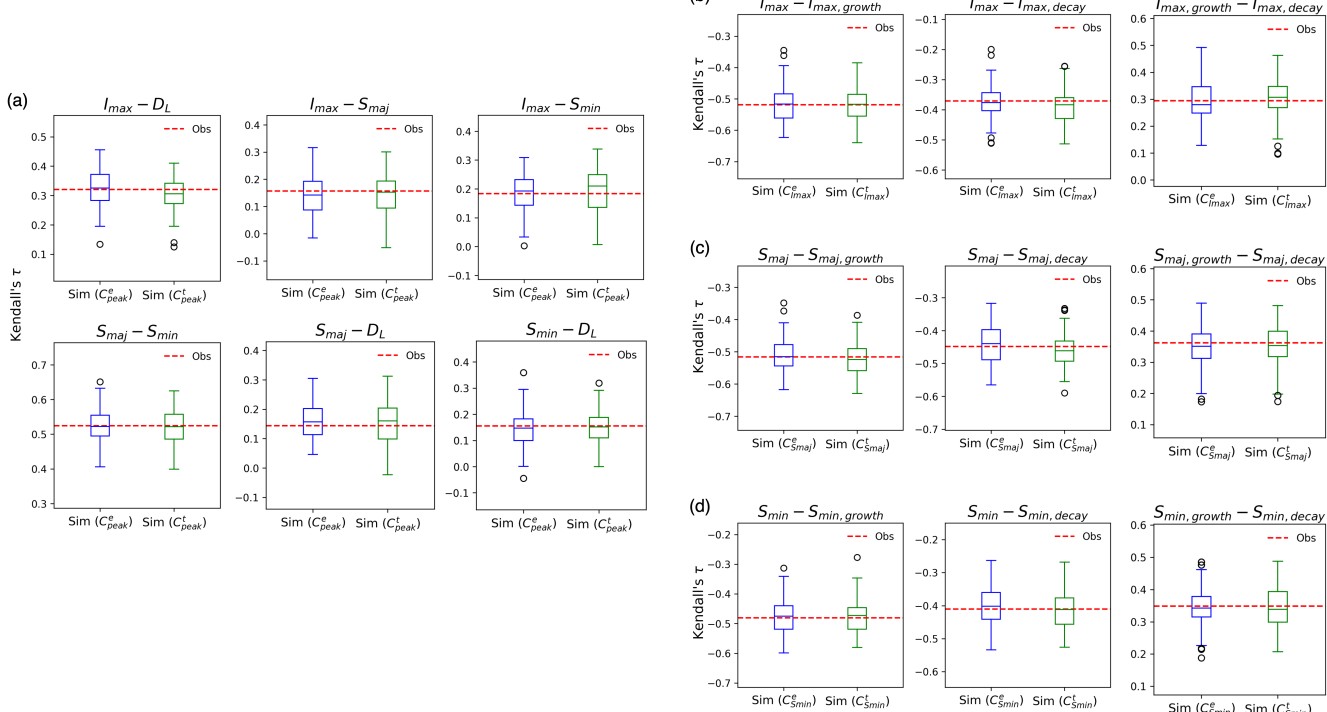

**Figure 13.** Comparisons of Kendall's $\tau$ correlations amongst cell properties, obtained from the observations (red dash line) and from simulations (blue boxplots: empirical quantile functions, and green boxplots: theoretical quantile function). Figure (a) are results between each pair of peak properties and lifespan; and Figure (b) are results between given peak properties and the corresponding growth and decay rates.

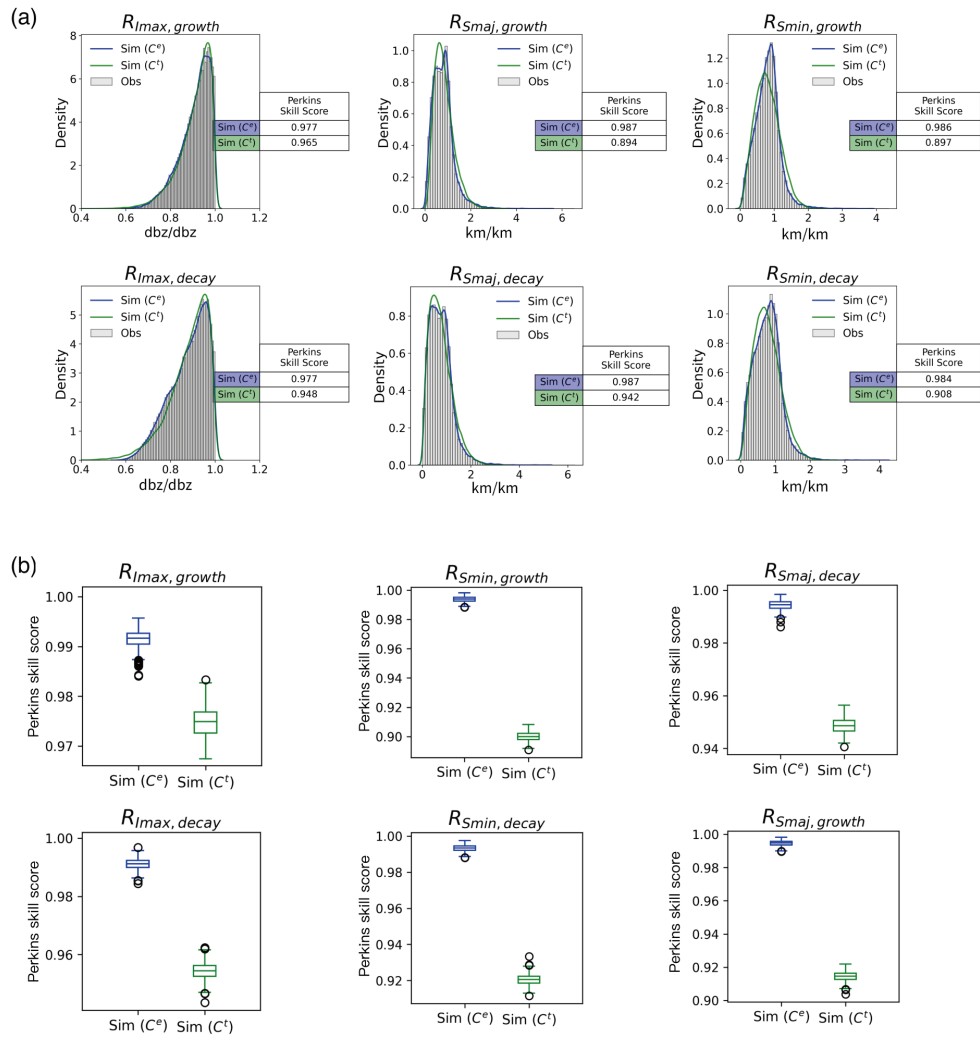

**Figure 14.** Comparisons of marginal distributions for observed and simulated growth and decay rates of selected cell properties (from left to right, $C_{Imax}$, $C_{Smaj}$ and $C_{Smin}$): (a) marginal distributions of an arbitrary ensemble member, and (b) distributions of Perkins skill scores derived from 100 ensemble members.

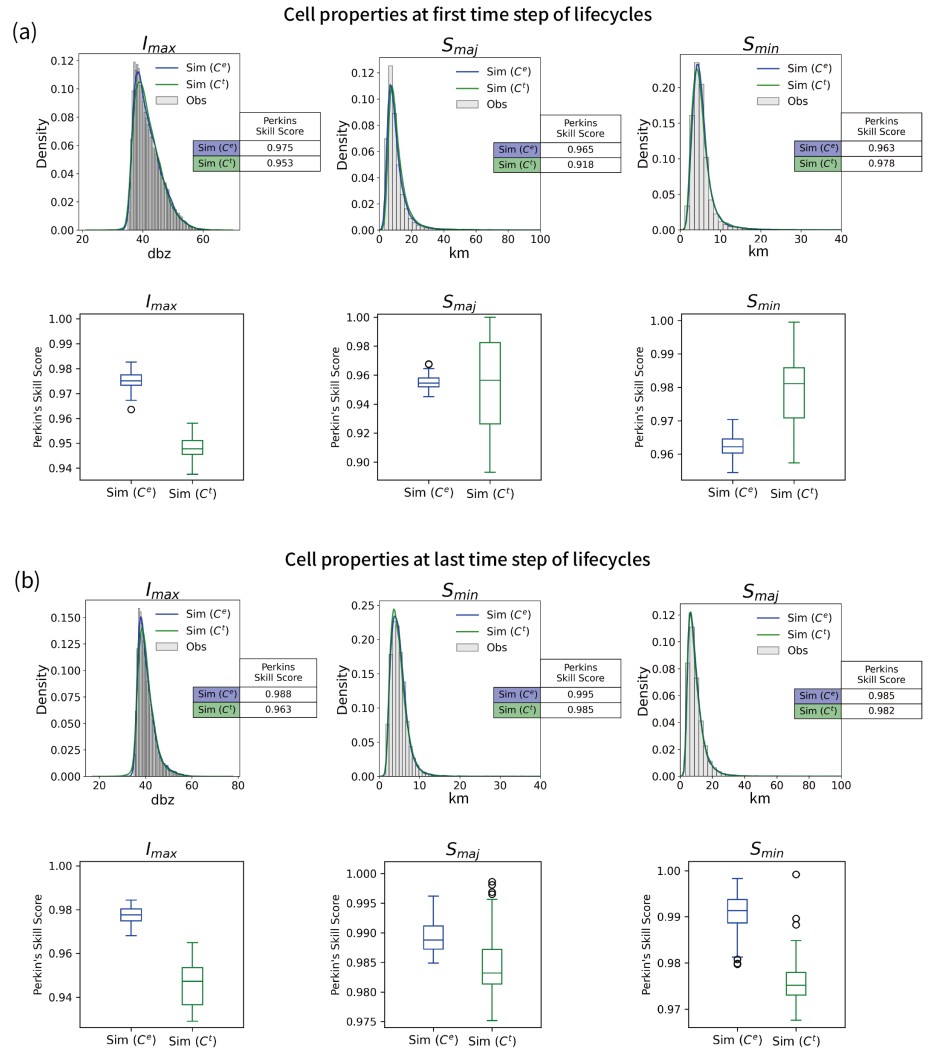

**Figure 15.** Distributions of observed and simulated properties for the (a) first and (b) last time step of the convective lifecycles.

## 5 Conclusions

This work proposes an algorithm that models the lifecycle of convective cells, explicitly accounting for the inter-dependence between cell properties and the corresponding temporal evolution using copulas. This algorithm enables stochastic generation

of samples of cell lifecycles, preserving not only the observed individual cell properties but also their dependence structures. A summary of the key findings throughout the development process is provided below:

- The proposed conceptual model and the generation algorithm effectively capture not only the cell properties at peaks but also their temporal variation throughout the cell's lifespan.

- Vine copulas are effective in modelling the complex high-dimensional inter-dependence amongst convective cell properties. Specifically, both the observed dependence structures amongst peak properties and between given peak properties and the corresponding growth and decay rates can be satisfactorily reproduced.

- While the correlations between cell properties have been widely discussed and modelled in the literature, this work demonstrates that the dependence between given properties and their temporal variations is not negligible and is useful in inferring cell properties at any stage within a lifecycle.

- In terms of sampling methods, the use of either empirical or theoretical quantile functions yields satisfactory results. The empirical conversion appears to slightly better preserve the observed distributions and dependence structures, whilst the theoretical conversion can simulate properties not directly observed in the dataset. Notably, in scenarios with large data sample sizes, such as those examined in this paper, the difference between these two functions is minimal. However, in cases with smaller sample sizes, the theoretical quantile function may be preferred to ensure a better inference of population features.

Beyond preserving convective cell properties, the proposed algorithm is methodologically generic. The use of vine copulas makes flexible the modelling of inter-dependence amongst selected properties, facilitating the incorporation of additional variables in the future work. With radar technology advancements, these may include variables derived from dual-pol data or 3-dimensional radar images (Rigo and Carmen Llasat, 2016; Cheng et al., 2024).

Furthermore, future work could expand the algorithm to account for more complex behaviours of convective cells, such as merging or splitting processes (Handwerker, 2002; del Moral et al., 2018). One possible approach would involve developing a two-stage framework, where the initial stage models the occurrence of cell merging or splitting, followed by a second stage that handles the evolution of cell properties post-interaction. This would enhance the algorithm's ability to capture the dynamic nature of convective systems.

In this work, we connected the proposed algorithm with the EXCELL model to generate spatially-distributed rainfall fields for sampled convective cells. Despite its simplicity, this linkage may effectively enhances the hydrological relevance of the proposed algorithm. Future research could further refine this connection using more complex models such as MultiEXCELL or HYCELL (Luini and Capsoni, 2011; Peleg and Morin, 2014), or by incorporating cell analog searching techniques (Shehu and Haberlandt, 2022).

In addition to linking with 'downstream' applications, the proposed algorithm could also be incorporated into 'upstream' applications. For example, future research could explore associating convective cell properties with large-scale atmospheric or climate variables. This would enable integration with climate models (e.g. Lucas-Picher et al. (2021b)) or statistically-based weather generators (e.g. Peleg and Morin (2014); Yiou (2014)), thereby facilitating the generation of convective storms within a climate context.

*Data availability.* The Nimrod radar data used in this work can be accessed via CEDA (Centre for Environmental Data Analysis).

*Author contributions.* Li-Pen Wang (LW) led the conceptualisation of research idea with support from Chien-Yu Tseng (CT) and Christian Onof (CO). CT led the model development and implementation with support from LW and CO. LW, CT and CO jointly designed the experiments to evaluate the result and prepared manuscript.

*Competing interests.* No competing interests are present.

*Acknowledgements.* The authors express their gratitude for the financial support received from two National Science and Technology Council research projects (NSTC 113-2625-M-002 -016- and 113-2923-M-002-001-MY4). In addition, they extend their appreciation to the Centre for Environmental Data Analysis for providing access to the NIMROD radar data produced by the Met Office.

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
