# Peer review of "Modelling convective cell lifecycles with a copula-based approach"

_EGUsphere, 2024_

## Author Comment (AC1)

[Figure]

Figure 1. (corresponding to Figure 4 in the original manuscript) Histogram of selected cell properties at peaks (a) and the corresponding growth/decay rates (b) fitted probability distributions.

[Figure]

Figure 2. (new figure will be added to the revised manuscript) An illustrative example of a convective cell lifecycle sampled from the proposed algorithm. Here, the EXCELL model is incorporated to further generate convective cells with spatially-distributed rainfall intensities at each time step based on the sampled properties.

[Figure]

Figure 3. (corresponding to Figure 5 in the original manuscript) Correlation analyses amongst selected cell properties: (a) between peak properties; (b)-(d) between each peak property and the associated growth and decay rates.

[Figure]

Figure 4. (new figure to be added in the Supplement S2) Visual inspection of the fitting results of parametric, non-parametric and mixed copula models.

[Figure]

Figure 5. Q-Q plots for the comparisons between the observed and simulated cell properties. Plots in (a), (b) and (c) correspond to Figures 8, 10 and 13 in the manuscript respectively.

[Figure]

Figure 6. (corresponding to Figure 9 in the original manuscript) A Comparison of dependence structures between observed and simulated cell lifecycle samples: (a) $I_{max, peak}$ vs. $D_L$ and (b) $S_{maj, peak}$ vs. $S_{min, peak}$. The left column in (a) and (b) presents results incorporating copula modelling (black crosses: observed, red dots: simulated), and the right column shows results without copula modelling (grey crosses: observed, red dots: simulated). The upper row displays dependence structure in the original variable space, while the lower row shows after applying the quantile transformation.

[Figure]

Figure 7. (corresponding to Figure 11 in the original manuscript) Comparisons of the dependence structure between observed (red round markers) and simulated properties obtained from an arbitrary ensemble member. From top to bottom, each row represents results derived from a specific copula model ($C_{peak}$, $C_{Imax}$, $C_{Smaj}$, and $C_{Smin}$).

Table 1. (corresponding to Table 1 in the original manuscript) Summary of key properties to conceptualise the lifecycle model and the corresponding optimal probability distribution and parameters.

| Property | Description | Fitted distribution | Distribution parameters | AIC |
|---|---|---|---|---|
| **Duration** | | | | |
| $D_{\mathrm{L}}$ | Total time duration of the cycle (5-min intervals) | Exponential | $\lambda{=}0.239, \gamma{=}3.000$ | 1.33E+05 |
| **Peak** | | | | |
| $I_{\mathrm{max,peak}}$ | Maximum intensity at peak (km) | Weibull | $\alpha{=}12.029, \beta{=}1.972, \gamma{=}35.528$ | 1.71E+05 |
| $S_{\mathrm{maj,peak}}$ | Major axis length at peak (km) | Loglogistic | $\alpha{=}9.807, \beta{=}2.261, \gamma{=}3.216$ | 1.93E+05 |
| $S_{\mathrm{min,peak}}$ | Minor axis length at peak (km) | Loglogistic | $\alpha{=}4.524, \beta{=}2.599, \gamma{=}1.915$ | 1.43E+05 |
| **Temporal variation** | | | | |
| $R_{I\mathrm{max,growth}}$ | The ratio of the initial to the peak maximum intensity (dbz/dbz) | Beta | $\alpha{=}13.373, \beta{=}1.329$ | -7.93E+04 |
| $R_{I\mathrm{max,decay}}$ | The ratio of the last to the peak maximum intensity (dbz/dbz) | Beta | $\alpha{=}10.636, \beta{=}1.44$ | -6.53E+04 |
| $R_{S\mathrm{maj,growth}}$ | The ratio of the initial to peak major axis length (km/km) | Gamma | $\alpha{=}0.255, \beta{=}3.074, \gamma{=}0$ | 2.75E+04 |
| $R_{S\mathrm{maj,decay}}$ | Ratio of the last to peak major axis length (km/km) | Weibull | $\alpha{=}0.830, \beta{=}1.654, \gamma{=}0.016$ | 2.95E+04 |
| $R_{S\mathrm{min,growth}}$ | Ratio of the initial to peak minor axis length (km/km) | Weibull | $\alpha{=}0.901, \beta{=}2.374, \gamma{=}0$ | 1.94E+04 |
| $R_{S\mathrm{min,decay}}$ | Ratio of the last to peak minor axis length (km/km) | Weibull | $\alpha{=}0.857, \beta{=}2.132, \gamma{=}0$ | 2.19E+04 |

Table 2. (corresponding to Table 2 in the original manuscript) Comparative evaluation of different copula models based on Akaike Information Criterion (AIC) and log-likelihood metrics.

| Vine-copula model | Control of Bivariate Family | AIC | log-likelihood |
|---|---|---|---|
| $C_{peak}$ | TLL | -37314.823 | 18912.031 |
| | Parametric | -32815.206 | 16416.603 |
| | **TLL and Parametric** | **-37362.077** | **18893.488** |
| $C_{Imax}$ | **TLL** | **-46064.656** | **23172.394** |
| | Parametric | -92409.25 | 46210.625 |
| | TLL and Parametric | -95783.66 | 47942.412 |
| $C_{Smaj}$ | **TLL** | **-40263.157** | **20269.259** |
| | Parametric | -29212.414 | 14611.207 |
| | TLL and Parametric | -40262.747 | 20269.104 |
| $C_{Smin}$ | **TLL** | **-42428.612** | **21353.235** |
| | Parametric | -39033.233 | 19521.616 |
| | TLL and Parametric | -42427.892 | 21352.897 |

---

## Author Comment (AC2)

Table 1:  Correlation analysis of convective cell properties' growth rate

| Kendall's Tau | $R_{Imax,growth}$ | $R_{Smaj,growth}$ | $R_{Smin,growth}$ |
|---|---|---|---|
| $R_{Imax,growth}$ | 1 | 0.197 | 0.197 |
| $R_{Smaj,growth}$ | 0.197 | 1 | 0.466 |
| $R_{Smin,growth}$ | 0.197 | 0.466 | 1 |

Table 2:  Correlation analysis of convective cell properties' decay rate

| Kendall's Tau | $R_{Imax,decay}$ | $R_{Smaj, decay}$ | $R_{Smin, decay}$ |
|---|---|---|---|
| $R_{Imax, decay}$ | 1 | 0.212 | 0.212 |
| $R_{Smaj, decay}$ | 0.212 | 1 | 0.526 |
| $R_{Smin, decay}$ | 0.212 | 0.526 | 1 |

[Figure]

Figure 1. (corresponding to Figure 5 in the original manuscript) Correlation analyses amongst selected cell properties: (a) between peak properties; (b)-(d) between each peak property and the associated growth and decay rates.

---

## Author Response (AR1)

**Author's Response**

We thank the Editor and the Reviewers for carefully reviewing our work and making constructive comments. We appreciate all the time and efforts they put in their thorough review. All the reviewer comments were considered in the revised manuscript. A summary of the main changes, and the detailed reply to each comment are given below.

Please note that the last reply (item 9) to the Review #2 has been updated. Both the original reply in the open discussion and the updated reply are provided.

**1. Main changes in the revised manuscript**

- A more detailed explanation of the event selection process is provided, including criteria based on the WaPUG guidelines (Section 2.3)
- A new table (Table 1 in the revised manuscript) is added, summarising the statistics of the selected (convective) storm events
- The visual representation of the convective cell lifecycle model is modified to reduce the level of confusion (Figure 3)
- The measure for individual model distribution selection has been replaced from the log-likelihood with AIC (Akaike Information Criterion) (Section 3.4.1), and the resulting AICs are summarised in Table 2.
- The visualisation of the observed and modelled correlation structures between cell properties has been updated. They are now displayed with standard uniform distributions of the variables over the interval [0,1] rather than with marginal distributions (Section 3.4.2 and Figure 5). As suggested by the Reviewer #1, the former is more commonly-seen in displaying correlation analysis results.
- The process of applying copula theory to model dependence has been elaborated, including its the two-step vine copula application process, as well as the use of `pyvinecopulib` Python package for model determination (Section 3.4.2).
- A supplement has been added, detailing the copula family selection and parameter estimation process (Supplement S1)
- The integration of the EXCELL model has been improved. These include an additional formula converting areal-averaged radar rainfall intensity to point rainfall intensity (Section 3.5), as well as an example of applying EXCELL model to generation a generated convective cell lifecycle using the EXCELL model (Figure 8)
- The visualisation of the observed and modelled dependence has been improved by including correlation structures displayed with standard uniform distributions (Figures 10-11).
- The order of figures in the result and discussion section and the corresponding descriptions have been updated (Figures 9-14).
- The future research and potential applications of the proposed algorithm have been extended. Particularly, the ability to model more complex convective processes, such as merging and splitting cells, in the future research has been highlighted (Section 5).

**2.    Referee #1**

**General opinion:**

1. The paper entitled "Modelling convective cell lifecycles with a copula-based approach" by Chien-Yu Tseng and co-authors proposes a new model for stochastic generation of convective rainfall cells. In my opinion the topic is relevant for the journal HESS, the proposed model is new and in general properly described, and the paper is well written.

   I nevertheless have two major concerns that should be addressed before publication. First, several steps of the statistical model are not sufficiently justified or explained, and depending on how they have been actually implemented they may be improper and therefore should be improved. Second, the simulation ——of actual rainfall fields based on rain cells properties is a very important application of the model and is mentioned in many places of the manuscript, but it is not at all illustrated in this paper. I strongly encourage the authors to implement this step and show the corresponding results in this paper.

   R/ We thank the Reviewer for the generally positive opinion about the proposed work. The major (and minor) concerns raised by the Reviewer RC1 will be addressed point by point below.

**Major concerns:**

·    *Unclear statistical model*

2. L 253-259: It seems that the marginal distribution models are selected based on the likelihood of each model in competition. However the different models may have different number of parameters, which makes a direct comparison of their likelihoods "unfair". A more standard model selection approach (e.g., AIC or BIC) should be used instead.

   R/ Thank you for your comment. You are correct that comparing models based solely on likelihood values can be misleading due to differences in model complexity. As you suggested, we have updated our model selection approach to use the Akaike Information Criterion (AIC) for a fairer comparison. We will update Table R1 and Figure R1 accordingly in the revised manuscript. However, as seen, this update does not change the current result of the best-fit distribution selection, so the remainder of the results remain unchanged.

   **Table R1.** (corresponding to Table 1 in the original manuscript) Summary of key properties to conceptualise the lifecycle model and the corresponding optimal probability distribution and parameters.

| Property | Description | Fitted distribution | Distribution parameters | AIC |
|---|---|---|---|---|
| **Duration** | | | | |
| $D_L$ | Total time duration of the cycle (5-min intervals) | Exponential | $\lambda=0.239, \gamma=3.000$ | 1.33E+05 |
| **Peak** | | | | |
| $I_{max,peak}$ | Maximum intensity at peak (km) | Weibull | $\alpha=12.029, \beta=1.972, \gamma=35.528$ | 1.71E+05 |
| $S_{maj,peak}$ | Major axis length at peak (km) | Loglogistic | $\alpha=9.807, \beta=2.261, \gamma=3.216$ | 1.93E+05 |
| $S_{min,peak}$ | Minor axis length at peak (km) | Loglogistic | $\alpha=4.524, \beta=2.599, \gamma=1.915$ | 1.43E+05 |
| **Temporal variation** | | | | |
| $R_{Imax,growth}$ | The ratio of the initial to the peak maximum intensity (dbz/dbz) | Beta | $\alpha=13.373, \beta=1.329$ | -7.93E+04 |
| $R_{Imax,decay}$ | The ratio of the last to the peak maximum intensity (dbz/dbz) | Beta | $\alpha=10.636, \beta=1.44$ | -6.53E+04 |
| $R_{Smaj,growth}$ | The ratio of the initial to peak major axis length (km/km) | Gamma | $\alpha=0.255, \beta=3.074, \gamma=0$ | 2.75E+04 |
| $R_{Smaj,decay}$ | Ratio of the last to peak major axis length (km/km) | Weibull | $\alpha=0.830, \beta=1.654, \gamma=0.016$ | 2.95E+04 |
| $R_{Smin,growth}$ | Ratio of the initial to peak minor axis length (km/km) | Weibull | $\alpha=0.901, \beta=2.374, \gamma=0$ | 1.94E+04 |
| $R_{Smin,decay}$ | Ratio of the last to peak minor axis length (km/km) | Weibull | $\alpha=0.857, \beta=2.132, \gamma=0$ | 2.19E+04 |

[Figure]

**Figure R1.** (corresponding to Figure 4 in the original manuscript) Histogram of selected cell properties at peaks (a) and the corresponding growth/decay rates (b) fitted probability distributions.

3. L294-301: Based on this description, it is unclear to me how the TLL copula has been selected/chosen, and why it has been preferred to parametric copulas. This should be explained in more details. For instance an AIC procedure is mentioned L285 and L287 for model selection, but how is the "model complexity term" computed in the case of a non-parametric copulas (i.e., the 2k term with k the number of model parameters and AIC=2k-2ln(L))?

In addition, how the TLL copulas are fitted and used must be described in more details. For instance is there any hyperparameter involved? If yes how is it selected?

R/ Thank you for your comment. In the AIC (Akaike Information Criterion) model selection process, the complexity term for non-parametric copulas like TLL is represented by effective degrees of freedom, which depend on the smoothing parameters, such as the bandwidth used in the kernel method. These effective degrees of freedom serve as a proxy for the number of parameters (k) in the AIC calculation. A more detailed explanation can be found in Nagler (2018).

Regarding TTL copulas fitting, it involves transforming the input data to uniform margins and using kernel density estimation to smooth the copula density. The smoothing parameters are crucial for determining the accuracy of the estimation. In this work, we employed the pyvinecopulib Python package to determine these parameters, where cross-validation is employed to optimise the parameter fitting process.

A detailed explanation of the copula model selection process will be provided in the Supplement S1, which includes the above description of TTL copula model fitting.

- *Simulation of rainfall fields*

4. The simulation of more realistic rainfall fields during convective events is a major selling point for the proposed method. This is mentioned in many places of the manuscript, but unfortunately the reader cannot find much details about the method that could be used to generate actual rainfall fields from rain cells properties. In addition, there is no illustration about how such rainfall fields would look like (neither in the form of rain maps, nor in terms of rainfall statistics).

R/ Thank you for your comment. It is indeed important to display the convective cells with spatially-distributed rainfall rates. An illustrative example is given in Figure R2, which presents the evolution of a simulated convective cell generated by our proposed algorithm throughout its lifecycle at six distinct timesteps. Here, the EXCELL model is used to further transform the sampled cell properties to cells with spatially-distributed rainfall intensities.

The upper part of Figure R2 showcases the three-dimensional structure of the cell at each timestep, with the maximum rainfall intensity (in mm/h) at the cell's peak and the major and minor axis lengths (in km) representing the cell's spatial extent. The lower part of Figure R2 presents a plan view of the simulated convective cell at each corresponding timestep. This view offers a clearer illustration of the changes in the cell's spatial spread over time, as generated by the EXCELL cell model. The concentric circles represent the rainfall intensity contours, with the innermost circle indicating the highest intensity.

As the cell evolves, the changes in both the three-dimensional structure in Fig R2 (a) and the contour patterns in Fig R2(b) reflect the growth and decay of the cell's spatial extent and rainfall intensity, as simulated by the proposed cell generator.

However, we would like to clarify that the proposed work is not intended to generate an entire rainfall field. Instead, it aims to generate spatially-distributed rainfall rates WITHIN each convective cell based on known cell properties (i.e. maximum intensity and major and minor extents for a given cell).

Figure R2, together with some explanation, will be added in the revised manuscript.

5. There is a brief mention and description of the EXCELL model that is envisioned to translate rain cells properties into rainfall fields (L 339-352), but many questions remain open. For instance: (1) are the rain cell advected, and if yes with which speed and direction? Should these parameters be linked to rain cell properties? (2) How do new rain cells enter the simulation domain? And in which stage of their development? (3) What is the rain cell density within the simulation domain? And what is the "birth rate" of new rain cells within the target area?

I invite the authors to address the question of how to simulate rainfall fields from the rain cell properties simulated by their current method, and to illustrate the results of this rainfall field simulation.

R/ Thank you for your comment. As mentioned in the previous reply, we would like to clarify that the proposed work focuses on sampling individual convective cell life-cycles rather than convective 'storms'. This means our focus is on modelling the evolution of individual convective cell properties. Parameters related to the advection of rain cells (e.g., motion speed and direction) and 'storm' sampling (e.g., cell density over the simulation domain, cell birth rates, and storm durations) are not considered in this work. It is, in fact, our ongoing work to develop a convective 'storm' generator that accounts for cell advection and evolution.

Regarding the EXCELL model, we use it as an exponential shape function to generate spatially-distributed rain rates within each convective cell. The reason for incorporating the EXCELL model is that the proposed algorithm only samples the selected cell properties throughout their lifespans. An additional model is required to translate these sampled properties into convective cells.

We will add more details regarding generating spatially-distributed rainrates within each convective cell in the revised manuscript.

[Figure]

**Figure R2**. (new figure will be added to the revised manuscript) An illustrative example of a convective cell lifecycle sampled from the proposed algorithm. Here, the EXCELL model is incorporated to further generate convective cells with spatially-distributed rainfall intensities at each time step based on the sampled properties.

Minor comments:

6. L13: "synthesise spatial fields of rainfall intensity for each cell" → I find this phrasing a bit misleading since I assume that the final rainfall intensity fields are made by the juxtaposition of intensities coming from several rain cells.

R/ Thank you for the comment. The phrase "synthesising spatial fields of rainfall intensity for each cell," describes the process in which our cell generator produces parameters for a spatial rain cell model. The sampled cell properties at a given time step, i.e. peak intensity, major and minor axis lengths, are used as input to a chosen spatial rain cell model (e.g., those following bi-exponential or bi-gaussian shape functions. Here, we chose EXCELL which follows the bi-exponential shape function) to simulate the spatial distribution of rainfall intensity associated with that individual convective cell. We will further this statement in the revised manuscript.

The final rainfall field would indeed involve the superposition of intensities from multiple cells, as you correctly noted. This is however part of the challenges of our ongoing work to incorporate this cell generator to a convective 'storm' generator. Here, because we only generate individual cell lifecycle, the superposition of multiple cells is not handled.

7. L80-83: Be more specific and better situate your work in relation to the above literature review. In particular, I have the impression that the proposed approach is an improvement of the step 2 of a point-process based rainfall model as mentioned at line 35. If this is the case it would be nice to state it clearly. In addition here would be a good place to briefly explain how the simulated rain cell properties would be used to generate rainfall fields, and how to deal with "side issues" such as advection, rain cell occurrence, etc.

R/ Thank you for the comment. We would like to clarify that our proposed cell generator aims to address limitations in existing storm generators, such as the one by McRobie et al. (2013), which often simplify the representation of convective cells. These simplifications include neglecting both the evolution of individual cells and the intercorrelation between cell properties. Our work addresses this challenge by focusing on the explicit simulation of individual convective cell lifecycles, capturing their dynamic evolution and property dependencies. The proposed cell generator could be implemented within the framework of existing rainfall generators to provide a more realistic representation of convective cells, instead of generating the convective 'storms'.

8. L97: It would be nice to show some data of these events, and in general of the dataset that will be used for application. Not necessarily in the main text, but maybe in supplementary material.

L127: Please provide a brief description of the WaPUG method.

R/ Thank you for your comments. The above two comments are relevant, so they are replied together here.

The WaPUG (now replaced by the CIWEM Urban Drainage Group (UDG), though still often referred to as WaPUG) has provided guidance to support best practices in urban drainage management in the UK since 1984. This includes guidelines for selecting storm events that comply with UK standards for the calibration and verification of urban drainage models. In this work, we specifically reference User Note 06 (Use of Rainfall Data from Flow Surveys) produced by WaPUG in 2009 (Gooch, 2009). Although a new rainfall modelling guide was published in 2016 (CIWEM, 2016), the principles for event selection remain similar. These principles include criteria for event durations, cumulative rainfall, instantaneous rainfall rates, and the quality of rainfall data.

For our study area, the general criteria are instantaneous rainfall rates greater than 5 mm/h and cumulative rainfall greater than 5 mm, which typically ensures effective rainfall and subsequent runoff. However, since our focus is on convective cell modelling, we specifically chose events between May and July, filtering out those without any 5-min rainfall intensity greater than 5.6 mm/h (equivalent to 35 dBZ—a threshold commonly used to identify convective regions according to the Marshall-Palmer relationship) and those with durations shorter than 15 minutes. We also excluded events with consecutive periods of missing radar data, resulting in a total of 165 events.

Together with the summary of selected storm events (that mentioned in the previous reply), we will provide a more detailed description of event selection in the Section 2.3 of the revised manuscript, including WaPUG standard and our event selection criteria.

9. L138-139: How to deal with rain cells with multiple cores as well as with cells splitting and merging seems an interesting future work. This may be mentioned in conclusion/perspectives.

R/ Thank you for the comment. The suggested statement will be added to the conclusions section of the revised manuscript.

10. L148-149: "preserve the observed statistical properties and inter-dependence of convective cells." → Said like this I have the impression that the proposed method models dependencies between rain cells (which if I understood well is not the case). Maybe a word is missing? → "inter-dependence of convective cells properties". Otherwise please rephrase.

R/ Thank you for the comment. The statement will be rephrased to:

'preserve the observed statistical properties and inter-dependence of convective cell properties.'

11. Sect 3.3 (starting L230): Please be more specific in the description of how the convective cell lifecycles are modeled. For instance: does the peak always occurs at Lifespan/2? Is the peak state "instantaneous" or does it last for a given duration? (if it is instantaneous please modify Fig. 3 accordingly). Are growth and decay linear? (at first I was sure that they were, but the dashed lines in Fig. 3 made me doubt)

R/ Thank you for the comment. Please find below for the clarification of our modelling of convective cell lifecycles:

- Peak Timing: To simplify the model, we assume that the peak intensity occurs at the midpoint of the cell's lifespan.

- Peak Duration: The peak state is considered instantaneous, occurring only within a single 5-min time step (based on the temporal resolution of our input radar data). We will adjust Figure 3 to better its 'instantaneous' setting in the revised manuscript.

- Growth and Decay: We employ linear growth and decay phases in our method. However, the dashed lines in Figure 3 are intended to illustrate that in reality, the extracted lifecycles may exhibit slightly more complex behaviour that deviates from the linear path during these phases.

12. L255-259: Implement a standard model selection approach (e.g., AIC or BIC) or justify why the log-Likelihood would be enough to select the best model.

R/ Thank you for your comment. As mentioned in one of the previous replies, it is not ideal to use the log-likelihood alone to determine the best model since it overlooks model complexity. Therefore, we have switched to using AIC for model selection. In summary, maximum likelihood estimation is employed for model parameters, and AIC is used to determine the best model. The statement will be modified accordingly in the revised manuscript.

13. L265-266: "do not adhere to a Gaussian distributions" → Do you mean "do not follow Gaussian distributions"?

R/ Thank you for the comment. The statement will be modified accordingly in the revised manuscript.

14. L266: "resolving these dependencies analytically is practically unfeasible" → I do not understand this statement very well, and if I try to guess what it means I disagree. The fact that the rain cell properties do not follow Gaussian distributions do not impede an analytical (do you mean parametric?) modeling. This is even the main interest of the copulas approach that you are using afterwards (if parametric copulas are used).

R/ Thank you for the comment. While analytical solutions exist for modelling dependencies between parameters following Gaussian or similar distributions, these approaches become unwieldy and inflexible when dealing with higher-dimensional, non-Gaussian relationships. In our case, the observed rain cell properties exhibit complex dependencies that cannot be adequately represented by assuming simple multivariate distributions.

The strength of the copula approach lies in its ability to decouple the modelling of the dependence structure from that of the marginal distributions of individual parameters. This allows us to capture the complex interrelationships between rain cell properties, even when they do not conform to standard multivariate distributions. Therefore, we utilise copula theory to construct a more flexible and accurate model of the dependencies inherent in our data, which would be highly challenging to achieve through traditional analytical methods. We will revise the phrasing in the manuscript.

15. L267: "utilise the theory of copulas to numerically model" : I think this is misleading about the copula approach (see my previous comment). Please rephrase.

R/ Thank you for the comment. We will revise the phrasing in the manuscript.

16. Figure 5 (and its description L269-277): the bivariate dependence plots are unreadable and not very informative about the dependence structure between variables (because the scatter-plots are dominated by the marginal distributions of the variables). Please replace the scatter-plots by empirical densities to improve readability (this comment is also valid for Fig.9 and 11). In addition instead of showing the correlation between "raw" variables I would rather transform the data using the parametric distributions inferred in Sect 3.4.1, and plot the dependencies between transformed data (i.e., in the [0,1]x[0,1] square) to align with the framework of copulas. To make my comment more clear: I propose to show the empirical copulas instead of the correlations between variables.

R/ Thank you for the comment. We agree that visualising the dependence structure between variables in the copula space (the [0,1]x[0,1] square) is more aligned with the proposed methodology. To address this, we have transformed the data using the pseudo-observation method. Pseudo-observations are obtained by applying the probability integral transform to the original data using the fitted marginal distributions, resulting in uniformly distributed values on the interval [0,1]. This transformation allows us to examine the dependence structure in the copula space, independently of the marginal distributions. The revised figure is provided below as Figure R3. We will revise the figure in the manuscript.

[Figure]

**Figure R3.** (corresponding to Figure 5 in the original manuscript) Correlation analyses amongst selected cell properties: (a) between peak properties; (b)-(d) between each peak property and the associated growth and decay rates.

17. L295-296: "Based on our analysis […] appears to be the most suitable model". This is unclear what this analysis is. Please be more specific, and in particular explain in details why you prefer TLL instead of parametric copulas.

R/ Thank you for the comment. It is indeed relevant to provide more details on our selection process of copula models. Together with one of the previous replies regarding TTL model fitting, we will provide a better explanation of our selection process in the Supplement S1. Please find below our reply regarding our selection process.

In our selection process, we evaluated three distinct fitting strategies for our vine-copula models. These are TLL (non-parametric), parametric and combined TLL and parametric strategies. These strategies differ in how they select the bivariate copula families for each edge of the vine structure:

- TLL (Non-parametric): This approach uses the non-parametric transformation local likelihood kernel estimator (TLL) family to estimate the bivariate dependencies.

- Parametric: This strategy fits a range of parametric copula families (Gaussian, Student's t, Clayton, Gumbel, Frank, Joe, BB1, BB6, BB7, BB8) and selects the best-fitting one based on a goodness-of-fit criterion.

- Combined TLL and Parametric: This strategy combines both TLL and parametric members.

The fitting results are summarised in Table R2, where the bold text highlights the vine-copula model selected as the optimal ones for each strategy. However, as can be seen, while the combined TLL and Parametric strategy initially appeared to be optimal for the CSmaj model based on AIC and log-likelihood values, the differences observed were abnormally high compared to other models. Thus, we further perform a visual examination of the simulations.

As shown in Figure R4, compared to the Combined strategy, the TLL (non-parametric) strategy provides a better visual match between the observed and simulated data for CSmaj, particularly in capturing the tail dependencies. The combined and (purely) parametric strategies exhibit a poorer fit, particularly noticeable in capturing the upper and lower tail dependencies. We think this might be due to numerical issues during the fitting process for the parametric models when applied to this specific variable.

Therefore, despite the initial results based on AIC, we have chosen the TLL model for CSmaj over a combined model. This decision prioritises the clear visual agreement between the observed and simulated dependencies, ensuring a more reliable representation of the underlying data structure.

**Table R2.** (corresponding to Table 2 in the original manuscript) Comparative evaluation of different copula models based on Akaike Information Criterion (AIC) and log-likelihood metrics.

| Vine-copula model | Control of Bivariate Family | AIC | log-likelihood |
|---|---|---|---|
| $C_{peak}$ | TLL | -37314.823 | 18912.031 |
| | Parametric | -32815.206 | 16416.603 |
| | **Parametric and TLL** | **-37362.077** | **18893.488** |
| $C_{Imax}$ | **TLL** | **-40263.157** | **20269.259** |
| | Parametric | -29212.414 | 14611.207 |
| | Parametric and TLL | -40262.747 | 20269.104 |
| $C_{Smaj}$ | **TLL** | **-46064.656** | **23172.394** |

| | | | |
|---|---|---|---|
| | Parametric | -92409.25 | 46210.625 |
| | Parametric and TLL | -95783.66 | 47942.412 |
| | **TLL** | **-42428.612** | **21353.235** |
| $C_{Smin}$ | Parametric | -39033.233 | 19521.616 |
| | TLL | -37314.823 | 18912.031 |

[Figure]

**Figure R4.** (new figure to be added in the Supplement S1) Visual inspection of the fitting results of parametric, non-parametric and mixed copula models.

18. L294-301: I'm very confused about what you are doing here. Please provide more details, and if possible with references. You mention both parametric and non-parametric copulas, this is confusing.

R/ Thank you for the comment. The lines 294-301 do lack clarity regarding the hierarchical nature of vine copula fitting. We will revise this section to explicitly explain that the selection of optimal bivariate copula families is performed at each tree level (Tree1, Tree2, etc.) of the vine structure, as illustrated in Table 2 in the original manuscript.

As noted in our previous response, we explored three distinct fitting strategies for this selection process. These strategies differ in the types of bivariate copula families considered for each paired variable within a given tree level: purely non-parametric, purely parametric, or a combination of both, which can refer to the comparative evaluation in Table R2 in the previous response. This aspect will be incorporated into the revised manuscript to provide a more comprehensive and transparent description of our model fitting methodology.

19. L371 and Fig. 8, Fig. 10: Could you add q-q plots in order to better see which part of the pdf is well (or poorly) simulated?

R/ Thank you for the comment. For your reference, we will include the q-q plots below in Figure 5. The blue points represent the empirical transformation, while the green points represent the theoretical transformation. Here, Figures R5 (a), (b), and (c) correspond to Figures 8, 10, and 13 in the original manuscript, respectively.

[Figure]

**Figure R5.** Q-Q plots for the comparisons between the observed and simulated cell properties. Plots in (a), (b) and (c) correspond to Figures 8, 10 and 13 in the manuscript respectively.

20. L385: "sample unseen properties values" → unobserved properties?

R/ Thank you for the comment. We will revise the phrasing in the manuscript.

21. L433: typo in "distributions"

R/ Thank you for the comment. The typo will be corrected in the revised manuscript.

22. Caption of Fig. 8: Remind what is Ce and Ct, and possibly also what are the cell properties.

R/ Thank you for the comment. We will provide a brief explanation of Ce and Ct, as well as the cell properties, in the revised manuscript.

23. Fig. 9 and 11: use empirical densities instead of scatter-plots. In contrast with Fig. 5, I think that for these two figures (Fig 9 and 11) showing correlations is fine since one want to evaluate the combined effects of the marginal distribution and the dependencies encoded by the copulas.

R/ Thank you for the comment. Relevant to one of the previous replies, Figures 9 and 11 in the original manuscript will be revised as Figures R6 and R7 repsectively as below.

[Figure]

**Figure R6.** (corresponding to Figure 9 in the original manuscript) A Comparison of dependence structures between observed and simulated cell lifecycle samples: (a) Imax, peak vs. DL and (b) Smaj, peak vs. Smin, peak. The left column in (a) and (b) presents results incorporating copula modelling (black crosses: observed, red dots: simulated), and the right column shows results without copula modelling (grey crosses: observed, red dots: simulated). The upper row displays dependence structure in the original variable space, while the lower row shows after applying the quantile transformation.

[Figure]

**Figure R7.** (corresponding to Figure 11 in the original manuscript) Comparisons of the dependence structure between observed (red round markers) and simulated properties obtained from an arbitrary ensemble member. From top to bottom, each row represents results derived from a specific copula model (Cpeak, CImax, CSmaj ,and CSmin).

24. L452: "can infer properties" → can simulate properties

R/ Thank you for the comment. We will revise the phrasing in the manuscript.

**3.    Referee #2**

1.  In my opinion, this paper tackles important scientific questions relevant to Hydrology and Earth System Sciences (HESS) by focusing on the modeling of convective cell lifecycles and their properties, which are crucial for understanding precipitation patterns and hydrological processes. To achieve this, the paper introduces a novel algorithm utilizing vine copulas to model convective cell lifecycles and integrates the EXCELL model for generating spatially-distributed rainfall intensities. The methods are clearly explained, and the document is well-structured and easy to follow. However, in my view,  some  questions/issues  need to be addressed.

    R/ We thank the Reviewer for the generally positive opinion about the proposed work. The comments the Reviewer raised will be addressed point by point below.

**Specific comments:**

2.  According to the introduction, a key motivation for this work is the challenge of integrating convective storm modeling into generators, especially in simulating the evolution of convective cells (Line 70), which is crucial for accurately representing extreme rainfall. Thus, an algorithm that explicitly models cell evolution over time is presented (Lines 81-83). However, in the conclusion section, the potential integration of the results into rainfall generators is only briefly mentioned in the last two lines.

    I encourage the authors to explain in more detail how their stochastic cell evolution generation method can be integrated into the model structure of rainfall generators that account for the spatial-temporal evolution of rainfall fields. Additionally, since "forecasting tools" are mentioned as another potential application in which the presented methodology can provide added values (Lines 13-14), I invite the authors to provide more details in the conclusion section about how this methodology could be integrated into such tools.

    R/ Thank you for your comment. For existing (spatial-temporal) rainfall generators or object-based rainfall nowcasting tools, it is widely seen that the evolution of the (convective) cells is not accounted for. For example, in the generator proposed by McRobie et al. (2013),the  evolution of the sampled storm cells' properties was not modelled. Similarly,  the nowcasting model proposed by Rossi et al. (2015) only conducted probabilistic forecasting of the positions of convective cells.

    The methodologically generic design of the proposed algorithm allows it to be incorporated into these existing models. For example, the incorporation with McRobie's rainfall generator is rather straightforward. Instead of sampling cells with properties that do not change with time, we actually sample cell lifecycles. That means, the properties (e.g. peak intensity and spatial extents) of a given cell will change as it moves. For nowcasting, the proposed algorithm can be useful in several aspects. One of them is to help predict cell lifespans conditioned on current property values. Specifically, based on the dependence (copula) model and the conditional sampling method proposed here, they can be used, in real time, to sample the distribution of the lifespans of a given cell conditioned on known cell extents and intensities. This is of great interest to operational applications.

    The description of the integration with existing models will be elaborated in the conclusion section.

3.  In Line 245, can you be more specific about what a multi-peak pattern is? What happens when, during a very long growth duration period, there is a "local maximum"? For example, when studying Imax, if you come across the following time series between the first time step and the peak: (40, 43, 46, 44, 49, 52, and 54 dBZ), there is a clear growth tendency interrupted in the 4th time step (44 dBZ). Is this case discarded?

R/ Thank you for your comment. The multi-peak pattern is indeed a relevant issue. In the proposed work, after retrieving individual cell lifecycles, a pattern with variations (or fluctuations) in property values can be commonly observed, as illustrated with the grey dashed line in Figure 3 of the original manuscript. This is similar to the example made by the Reviewer here. As explained in Section 3.3, we proposed a conceptual model to characterise cell lifecycles with properties at three key time steps. These are:

- First: The initiation of the cell's life.

- Maximum Imax: The point of peak intensity.

- Last: The final stage before the cell dissipates.

We then approximated the growth and decay of the cell properties as linear processes. This is indeed a simplification, but enables us to capture the essence of the cell's growth and decay pattern.

Specifically to the example time series (indicating Imax values from the first to the peak time steps) given the Reviewer, the proposed model will employ only the 40 (first) and the 54 (peak) dBZ values and treat it as a linear growth. Therefore, the "multi-peak" pattern is actually distilled into a single-peak pattern based on our defined principle.

4. From Figure 3 and Section 3.5, one can assume that if the cell lifecycle duration is D, the time between the first time step and the peak is D/2. If so, in which time step do you place the peak when the total duration involves, for example, 4 time steps, 6, 8, or 10?

R/ Thank you for your comment. The current method randomly assigns the peak to one of the two middle time steps when the duration involves an even number of time steps. For the case of 4 time steps, for example, the peak will be assigned at either time step 2 or 3 randomly.

5. The correlation analysis between growth and decay rates for each property was performed (Fig. 5). But what about the possible relationship between the Smaj-Smin growth rates as well as Imax-Smaj and Smin-Imax growth rates? Was that included in the analysis?

R/ Thank you for your comment. The point raised regarding the potential relationships between the growth rates of Smaj-Smin, Imax-Smaj, and Smin-Imax is indeed an interesting aspect worth further exploration. An analysis is conducted here to investigate the dependence of growth and decay rates. respectively, amongst selected properties. As shown in Tables R3 and R4, indeed the correlation coefficients of Smaj-Smin growth and decay rates are not low and should not be neglected. However, we didn't explicitly consider these two dependencies here for one main reason –model complexity.

Our research methodology is designed based on our understanding of the natural evolution process of convective cells, aiming to capture the most fundamental and crucial relationships in the lifecycle of convective cells. In the current two-stage setup, involving the aforementioned dependence of Smaj-Smin growth or decay rates may lead to a 6D copula. This would largely increase the complexity for copula model selection and construction. In addition, as illustrated in Figure 13 of the original manuscript, the current setup can accurately infer cell properties (Imax, Smaj, and Smin) at the beginning and end of lifecycle samples. This suggests the proposed algorithm can fulfil our purpose to effectively capture the essential dynamics of convective cell lifecycles without accounting for additional dependence.

Nonetheless, we recognise the potential value of exploring the additional relationships suggested. We will include this as a potential technical improvement for the further work.

**Table R3.** Correlation analysis of convective cell properties' growth rate

|  | Imax_GrowthRate | Smaj_GrowthRate | Smin_GrowthRate |
|---|---|---|---|
| Imax_GrowthRate | 1 | 0.197 | 0.197 |
| Smaj_GrowthRate | 0.197 | 1 | 0.466 |
| Smin_GrowthRate | 0.197 | 0.466 | 1 |

**Table R4.** Correlation analysis of convective cell properties' decay rate

|  | Imax_DecayRate | Smaj_DecayRate | Smin_DecayRate |
|---|---|---|---|
| Imax_DecayRate | 1 | 0.212 | 0.212 |
| Smaj_DecayRate | 0.212 | 1 | 0.526 |
| Smin_DecayRate | 0.212 | 0.526 | 1 |

6. Why in figure 5a there are some kendall values are in red and others in black?

R/ Thank you for your comment. The values labeled in red represent correlations larger than 0.2, which matches the threshold used to identify correlated variables in our analysis.

7. Why the plot of the distribution of Imax is different when comparing the one given in 5a with the one given in 5b?

R/ Thank you for your comment and for pointing out this difference. The scale of the y-axis of the distributions of Imax in Figures 5a and 5b, was automatically generated and was incorrectly assigned with scale for other properties.

To address this editing error and to adopt a more commonly-used way to visualise the dependence structure between variables in the copula analysis, we change to present the dependence structure in the copula space (i.e. over [0,1] x [0,1] scales) using pseudo-observations. This involves transforming the data using the probability integral transform with the fitted marginal distributions, resulting in uniformly distributed values on the interval [0,1]. This transformation allows for a clearer examination of the dependence structure within the copula space, independent of the marginal distributions. The revised figure is provided below as Figure R8. We will revise the figure in the manuscript.

[Figure]

**Figure R8.** (corresponding to Figure 5 in the original manuscript) Correlation analyses amongst selected cell properties: (a) between peak properties; (b)-(d) between each peak property and the associated growth and decay rates.

8. In Line 284 when the authors say ''we employ the Akaike information criterion (AIC) to determine the optimal vine structure for cell properties'' and then in Line 288 '' A 4-dimensional (4D) 2-3-1-4 D-vine copula is used to model the dependence amongst cell duration and peak properties'' it is not possible to see how different is using for example the current 2-3-1-4 D-vine copula with regards to 3-2-1-4. Thus, it is recommended to create a table with the different D-Vine copulas analysed to support the final D-Vine copula selection.

R/ Thank you for your comment. We acknowledge that a clearer justification for the selected vine structure is needed. While listing all considered structures and their AIC values might be excessive, we can provide a concise explanation based on the model selection procedure and the dependence structure amongst variables.

The selection of the optimal vine structure utilises a sequential method, detailed in the Section 3.1 and Algorithm 3.1 of Dißmann et al. (2013) This method prioritises stronger dependencies, measured by Kendall's tau, within the general framework of R-vines. In our study, we implemented this method using the `pyvinecopulib` package in Python, which identified the 2-3-1-4 D-vine structure as optimal, indicating that the dependence pattern is best captured by this particular D-vine structure.

The result is further supported by the observed Kendall's tau values. The strongest pairwise dependencies are found between Variables 2 and 3 ($\tau = 0.53$), Variables 1 and 4 ($\tau = 0.45$), and Variables 3 and 1 ($\tau = 0.28$). The 2-3-1-4 structure strategically places these pairs close within the vine, effectively reflecting the data's inherent dependence pattern.

9. From the results section, in my opinion is confusing to place Figure 9, between Figure 8 and 10. From what I understand, first you want to show that marginal distributions and dependence

structures are preserved after simulation. Therefore, I see more appropriate to put Figure 8, Figure 10, Figure 11 and Figure 12 one after each other and put the current Figure 9 after these four figures. This might imply also a bit of change in the structure of the paragraphs.

*(THE ORIGINAL REPLY R/ Thank you for your comment. We agree that the location of Figure 9 can be indeed confusing after a detailed review of the manuscript's structure. As pointed out, Figure 9 demonstrates the differences in correlation when a copula method is incorporated or not. To better emphasise the importance of utilising copulas, we will adjust the figure order by moving Figure 9 to a place before current Figure. 8. This will highlight the impact of the copula method on preserving marginal distributions and dependence structures in the subsequent figures.)*

R/ Thank you for your comment. We agree that the location of Figure 9 can be indeed confusing after a detailed review of the manuscript's structure. However, after several adjustment trials to the manuscript, we have decided to maintain the related position of Figure 9 (Fig.10 in the revised manuscript) but re-organised the order of several figures after this figure. To address your concerns, we have enhanced the connections and improved the flow between the figures in the corresponding paragraphs.

Specifically, we first discuss the statistical characteristics of individual cell properties. We then introduce a figure that demonstrates the importance of the copula method. This figure emphasises the necessity of using copulas when modelling correlations, as well as serves a good connection between individual and multivariate distributions. We then proceed with a detailed examination of correlations between properties. Finally, we compare the effects of different marginal structure conversion methods. We believe that this updated structure can provide you with a clearer understanding of our analytical process and the relationships between the figures.

**References**

Dißmann, J., Brechmann, E., Czado, C., and Kurowicka, D.: Selecting and Estimating Regular Vine Copulae and Application to Financial Returns, Computational Statistics & Data Analysis, 59, 52–69, https://doi.org/10.1016/j.csda.2012.08.010, 2013.

McRobie, F. H., Wang, L. P., Onof, C., & Kenney, S. (2013). A spatial-temporal rainfall generator for urban drainage design. Water Science and Technology, 68(1), 240-249.

Nagler, T. (2018). kdecopula: An R Package for the Kernel Estimation of Bivariate Copula Densities. Journal of Statistical Software, 84(7), 1–22. https://doi.org/10.18637/jss.v084.i07

Rossi, P. J., Chandrasekar, V., Hasu, V., & Moisseev, D. (2015). Kalman filtering–based probabilistic nowcasting of object-oriented tracked convective storms. Journal of Atmospheric and Oceanic Technology, 32(3), 461-477.

---

## Author Response (AR2)

**Modelling convective cell lifecycles with a copula-based approach**

Chien-Yu Tseng[1], Li-Pen Wang[1,2], and Christian Onof[2]

[1]National Taiwan University, Taipei, 106319, Taiwan
[2]Imperial College London, London, SW7 2AZ, United Kingdom

Correspondence: Li-Pen Wang (lpwang@ntu.edu.tw)

**Authors' Response**

**1. Referee #1**

1. I have reviewed the content thoroughly and find no suggestions for revision, as the current presentation effectively captures the intended insights with clarity and depth.

   R/ We appreciate the Reviewer's detailed follow-up review and positive feedback. We are glad that the updates satisfactorily address the points previously raised by the reviewer.

**2. Referee #2**

1. This is my second review of the paper "Modelling convective cell lifecycles with a copula-based approach" by Chien-Yu Tseng and co-authors. The authors addressed all my concerns in their response to my comments, and modified their manuscript accordingly. The only comment that has not been implemented in the paper is the generation of precipitation fields using the simulated cell properties, but the authors convincingly justified their choice of keeping this aspect for future work, and I agree with their point. So in my opinion the paper is now ready for publication.I have some last minor comments that came to my mind during my second reading of the paper, and that I'm listing below. I think they can be implemented by the authors at the type-setting stage if they find these comments relevant (the line numbers refer to the revised manuscript without tracked changes):

   R/ We appreciate the Reviewer's thorough second review and positive feedback. We are pleased that the revisions addressed the previous concerns. The minor comments the Reviewer raised will be addressed point by point below.

2. L19: often attributed severe convective systems => often attributed to severe convective systems.

   R/ Thank you for the comment. We have added 'to' to the sentence in the revised manuscript.

3. Figure 1: It would be nice to add a scale on the right panel. In addition I wonder if you should not replace the 75km buffer by a 200km buffer. I say that because 75km is not mentioned in the text, while 200km is. And in the current figure one could have the impression that the whole domain is not covered by the weather radars (there are white gaps in the left panel), which is fortunately not the case.

   R/ Thank you for your comment. The 50 km and 75 km buffers were initially intended to highlight that the centre of the study area was covered with radar data of relatively good quality. However, we agree that the 200 km buffer can better represent the operational coverage particularly for the study domain. The revised figure (see Fig. R1) now includes the 200 km buffer around the radar range, better aligning with the analysis context.

[Figure]

Fig R1. Pilot catchment, study domains and rainfall monitoring networks. Pilot catchment, study domains and rainfall monitoring networks.

4. Table 1: are the intensities referring to 5-min time steps? If yes it could be useful to specify it somewhere in the caption. I say that because the unit (mm/h) can be confusing.

   R/ Thank you for the comment. It is indeed essential to specify 'time interval' while showing intensity in mm/h. We have revised both the main text in L145 and the column names in Table 1 to "5-min Ave. Areal Peak" and "5-min Ave. Max. Peak.".

5. L287: It may be worthwhile to add a (relatively general) reference about copulas at this place.

   R/ Thank you for the comment. We have added a couple of general references about copulas (e.g., Genest and Favre, 2007; Jaworski et al., 2013; Czado, 2019; Tootoonchi et al., 2022) to the revised manuscript.

6. Figure 5: I think you could keep only the upper right part of each panel (the diagonal is not very informative).

   R/ Thank you for your comment. Indeed, the upper triangular part (excluding the main diagonal) of each panel provides the main message here, indicating inter-dependencies between cell properties under investigation. However, we believe that the diagonal part offers a valuable visual aid in interpreting the results of transforming the marginal distributions to a standard uniform distribution over the interval [0, 1]. The full correlation matrix ensures consistency with the expected correlations and aids in our comprehensive model assessment. Therefore, we choose to retain the diagonal in Figure 5.

**References**

Czado, C.: Analyzing Dependent Data with Vine Copulas: A Practical Guide With R, vol. 222 of Lecture Notes in Statistics, Springer International Publishing, ISBN 978-3-030-13784-7 978-3-030-13785-4, https://doi.org/10.1007/978-3-030-13785-4, 2019.

Genest, C. and Favre, A.-C.: Everything You Always Wanted to Know about Copula Modeling but Were Afraid to Ask, 12, 347–368, https://doi.org/10.1061/(ASCE)1084-0699(2007)12:4(347), 2007.

Jaworski, P., Durante, F., and Härdle, W. K., eds.: Copulae in Mathematical and Quantitative Finance: Proceedings of the Workshop Held in Cracow, 10-11 July 2012, vol. 213 of Lecture Notes in Statistics, Springer Berlin Heidelberg, ISBN 978-3-642-35406-9 978-3-642-35407- 6, https://doi.org/10.1007/978-3-642-35407-6, 2013.

Tootoonchi, F., Sadegh, M., Haerter, J. O., Räty, O., Grabs, T., and Teutschbein, C.: Copulas for Hydroclimatic Analysis: A Practice-oriented Overview, 9, e1579, https://doi.org/10.1002/wat2.1579, 2022.